



# Sea spray aerosol organic enrichment, water uptake and surface tension effects

Luke T. Cravigan[1], Marc D. Mallet[1,a], Petri Vaattovaara[2], Mike J. Harvey[3], Cliff S. Law[3,4], Robin L. Modini[5,b], Lynn M. Russell[5], Ed Stelcer[6,c], David D. Cohen[6], Greg Olsen[7], Karl Safi[7], Timothy J. Burrell[3], and Zoran Ristovski[1]

[1]International Laboratory for Air Quality and Health, CPME, Queensland University of Technology, Brisbane, Australia
[a]Now at Defence Science and Technology Group, Melbourne, Australia
[2]University of Eastern Finland, Kuopio, Finland
[3]National Institute of Water and Atmospheric Research, Wellington, New Zealand
[4]Department of Marine Sciences, University of Otago, Dunedin, NZ
[5]Scripps Institute of Oceanography, University of California, San Diego, La Jolla, California
[b]Now at Laboratory of Atmospheric Chemistry, Paul Scherrer Institute, 5232 Villigen PSI, Switzerland
[6]Centre for Accelerator Science, NSTLI, Australian Nuclear Science and Technology Organisation, Menai, NSW, Australia
[c]Deceased
[7]National Institute of Water and Atmospheric Research, Hamilton, New Zealand

**Correspondence:** Zoran Ristovski (z.ristovski@qut.edu.au)

**Abstract.** The aerosol driven radiative effects on marine low-level cloud represent a large uncertainty in climate simulations, in particular over the Southern Ocean, which is also an important region for sea spray aerosol production. Observations of sea spray aerosol organic enrichment and the resulting impact on water uptake over the remote southern hemisphere are scarce, and are therefore the region is under-represented in existing parameterisations. The Surface Ocean Aerosol Production (SOAP)

voyage was a 23 day voyage which sampled three phytoplankton blooms in the highly productive water of the Chatham Rise, east of New Zealand. In this study we examined the enrichment of organics to nascent sea spray aerosol and the modifications to sea spray aerosol water uptake using in-situ chamber measurements of seawater samples taken during the SOAP voyage.

Primary marine organics contributed up to 23% of the sea spray mass for particles with diameter less than approximately $1 \mu m$, and up to 87% of the particle volume in the Aitken mode. The composition of the organic fraction was consistent

throughout the voyage and was largely comprised of a polysaccharide-like component, characterised by very low alkane to hydroxyl concentration ratios of approximately 0.1 - 0.2. The enrichment of organics was compared to the output from the chlorophyll-a based sea spray aerosol parameterisation suggested by Gantt et al. (2011) and the OCEANFILMS models. OCEANFILMS improved on the representation of the organic fraction predicted using chlorophyll-a, in particular when the co-adsoprtion of polysaccharides was included, however the model still under predicted the proportion of polysaccharides by

an average of 33%.

Nascent sea spray aerosol hygroscopic growth factors averaged $1.93 \pm 0.08$, and did not decrease with increasing sea spray aerosol organic fractions. The observed hygroscopicity was greater than expected from the assumption of full solubility, particularly during the most productive phytoplankton bloom (B1), during which organic fractions were greater than approximately 0.4. The water uptake behaviour observed in this study is consistent with that observed for other measurements of phytoplank-





ton blooms, and was attributed to the surface partitioning of the organic components which leads to a decrease in particle surface tension and an increase in hygroscopicity. The compressed film model was used to estimate the influence of surface partitioning and the error in the modelled hygroscopicity was low only when the entire organic fraction was available to partition to the particle surface. The modelled sea spray aerosol hygroscopicity at high organic fractions was underestimated when

only a portion of the organic component was available to be partitioned to the surface. The findings from the SOAP voyage highlight the influence of biologically-sourced organics on sea spray aerosol composition, these data improve the capacity to parameterise sea spray aerosol organic enrichment and water uptake.

## 1   Introduction

Aerosol-cloud interactions represent a large uncertainty in modelled radiative forcing (Myhre et al., 2013), and in particular a

strong radiative bias has been associated with under representation of low level cloud over the Southern Ocean (Bodas-Salcedo et al., 2012; Protat et al., 2017). Sea spray aerosol (SSA) are an important contributor to cloud condensation nuclei (CCN), in particular over the Southern Ocean (Quinn et al., 2017; Cravigan et al., 2015; Fossum et al., 2018; Gras and Keywood, 2016). The phytoplankton derived organic fraction of nascent SSA has been shown to influence water uptake and CCN activity, however in some cases this relationship is not entirely predictable. For example the drivers of an observed positive correlation

between modelled SSA organic mass fraction and cloud droplet number concentrations over the high latitude Southern Ocean are unclear (McCoy et al., 2015). Models of SSA organic enrichment are largely based on chamber measurements which relate SSA and seawater composition. SSA generation chambers are often used because ambient SSA measurements are hindered by low concentrations and atmospheric processing (Cravigan et al., 2015; Frossard et al., 2014; Laskin et al., 2012; Shank et al., 2012). SSA measurements are scarce in the Southern Hemisphere (Cravigan et al., 2015), therefore parameterisations of SSA

organic enrichment have largely been developed based on Northern Hemisphere measurements. Further constraint of these relationships to observations for the Southern Hemisphere is required.

The inorganic composition of SSA has largely been assumed to mirror the inorganic composition of seawater (Lewis and Schwartz, 2004), however a few important exceptions have been identified. The enrichment of $Ca^{2+}$ in SSA, relative to seawater, has been identified in a number of ambient marine (Leck and Svensson, 2015; Sievering et al., 2004) and nascent SSA

chamber studies (Cochran et al., 2016; Keene et al., 2007; Salter et al., 2016; Schwier et al., 2017). Cation enrichment has also been observed in the form of $Mg^{2+}$ and $K^+$ (Ault et al., 2013b; Schwier et al., 2017), and is also associated with deficits in $Cl^-$ (Ault et al., 2013a; Prather et al., 2013; Schwier et al., 2017). $Ca^{2+}$ has been observed to be enriched by up to 500% (Schwier et al., 2017), relative to seawater, with stronger enrichment for smaller diameter SSA. It is uncertain whether the ocean surface $Ca^{2+}$ is complexed with organic matter, or is the product of precipitation of $CaCO_3$ by certain phytoplankton groups, which

can be a product of photosynthetic reactions. Observations of organic carbon particles that contain inorganic cations (such as $Ca^{2+}$, $Mg^{2+}$ and $K^+$) but no $Cl^-$, has also been reported, suggesting that in some cases these species complex with organic material (Ault et al., 2013a).





Chamber observations of nascent SSA universally indicate the presence of an internally mixed organic component, with the organic contribution varying between studies from approximately 4% - 80% by volume (Fuentes et al., 2011; Schwier et al., 2015, 2017; Facchini et al., 2008; Bates et al., 2012; Modini et al., 2010a). The majority of observations show that the organic fraction increases with decreasing particle diameter (Facchini et al., 2008; Keene et al., 2007; Prather et al., 2013; Quinn et al.,
2014). Estimates of the SSA organic fraction based on water uptake methods are of the order of 5 - 37% (Fuentes et al., 2011; Modini et al., 2010a), and sit at the lower end of the all nascent SSA organic fraction measurements (Facchini et al., 2008; Keene et al., 2007; Prather et al., 2013). Exceptions include Quinn et al. (2014), who observed an organic volume fraction (OVF) of up to 0.8 using CCN measurements. Externally mixed organics have also been observed to comprise the majority of the number concentration in some studies (Collins et al., 2014, 2013; Prather et al., 2013). It should be noted that the externally
mixed organic fraction still contained inorganic ions, but was characterised by an absence of Cl (Ault et al., 2013a).

The organic fraction of SSA appears to be comprised of a volatile component which evaporates at approximately 150 - 200°C and comprises of the order of 10% of the SSA volume (Modini et al., 2010a; Quinn et al., 2014). An SSA non-volatile organic component has also been observed (Quinn et al., 2014), but was not seen in the South-West Pacific (Modini et al., 2010a). The organic volume fraction has been previously estimated by comparing the volatility of nascent SSA generated using natural
seawater samples with SSA generated from sea salt samples (Cravigan, 2015; Mallet et al., 2016; Modini et al., 2010a). This method assumes that the volatile organic component is the only cause for the difference in volatility. It has been subsequently noted that components of sea salt aerosol retain hydrates even when dried to very low RH (<5%), such as $MgCl_2$ and $CaCl_2$ (Rasmussen et al., 2017). Sea salt hydrates need to be taken into account when estimating the SSA OVF using volatility.

The SSA organic fraction has been associated with the phytoplankton decline due to bacterial grazing and viral infection,
which releases fatty acids and polysaccharides predisposed to SSA enrichment (O'Dowd et al., 2015). Consistent with this are observations that SSA organics are characterised by a large fraction (>50%) of hydroxyl functional groups, with smaller contributions from alkanes and smaller again from amines and carboxylic acid groups (Russell et al., 2010). The hydroxyl functional group is generally consistent with water soluble polysaccharides (Russell et al., 2010), however water insoluble lipopolysaccharides (LPS) have been observed in SSA (Ceburnis et al., 2008; Facchini et al., 2008; Claeys et al., 2010; Miyazaki et al.,
2010; Sciare et al., 2009) and may depress water solubility and enhance surface tension effects. Functional group measurements can be equated to molecular classes, for example higher alkane to hydroxyl ratios indicates less oxygenated and more lipid like organics. Chamber generated nascent SSA alkane to hydroxyl ratios averaging $0.34 \pm 0.21$ for non-productive waters and $0.93 \pm 0.41$ for productive waters have been reported (Frossard et al., 2014). The proportion of fatty acids, and therefore alkanes, is linked to phytoplankton production because the lifetime of fatty acids is relatively short, compared to polysaccharides .

Empirical relationships between the nascent SSA organic fraction and chlorophyll-a (Chl $a$) concentrations are most commonly used to estimate SSA organic enrichment (Gantt et al., 2011; Schwier et al., 2015; Fuentes et al., 2011). A lagged correlation between the peak in Chl $a$ of a phytoplankton bloom and organic enrichment of SSA has been observed (O'Dowd et al., 2015; Rinaldi et al., 2013). Although Chl $a$ may not represent the molecular processes that drive enrichment it indicates the spatial and temporal magnitude of phytoplankton whilst providing a proxy for organic compounds released. Chl $a$ is readily
available due to satellite retrievals (Rinaldi et al., 2013) and it has been shown that over monthly timescales correlates well with



SSA organic fraction (O'Dowd et al., 2015). Alternative models have been suggested which aim to better explain the molecular drivers of SSA organic enrichment. OCEANFILMS (Organic Compounds from Ecosystems to Aerosols: Natural Films and Interfaces via Langmuir Molecular Surfactants) is a model of SSA organic enrichment based on the seawater molecular composition (Burrows et al., 2014, 2016). Marine organics are broken up into classes and adsorption to a surface film (e.g. a bubble) is

driven by the Langmuir adsorption coefficient. The molecular classes are a labile lipid-like class, a semi-labile polysaccharide-like class, a protein class with intermediate ocean lifetimes, a long lived processed class and a humic-like mixture from deep up-welled water. The processed class composition and surface activity is poorly characterised, but this class represents the recalcitrant dissolved organic carbon resulting from biogeochemical ageing of other groups. The Langmuir adsorption coefficient is based on observations of reference molecules for each molecular class (Burrows et al., 2014). The surface coverage

representation is further extended by considering the interaction between polysaccharides and more surface active molecular classes, which results in co-adsorption of more soluble polysaccharides (Burrows et al., 2016). Only a small number of long term datasets are available to constrain SSA enrichment, in particular in the Southern Hemisphere (Quinn et al., 2015).

Chamber studies have largely indicated that SSA organics are soluble and follow the Zdanovskii, Stokes, and Robinson (ZSR) assumption (Stokes and Robinson, 1966), with HGFs for organically enriched SSA suppressed by 4-17% (Bates et al.,

2012; Fuentes et al., 2011; Modini et al., 2010a; Sellegri et al., 2006; Quinn et al., 2014; Schwier et al., 2015). Importantly, exceptions have been identified which indicate a role of surface tension on SSA water uptake (Ovadnevaite et al., 2011a; Collins et al., 2016; Forestieri et al., 2018). The suppression of surface tension has been identified as having a potential role in values of the hygroscopicity parameter ($\kappa$) observed during nascent SSA microcosm experiments, which were persistently high ($\kappa > 0.7$) even with high marine biological activity and high SSA organic fractions (Collins et al., 2016). Alternatives to

the assumption of full solubility for the organic component of internal carboxylic acid-salt mixtures have been suggested based on laboratory (Ruehl and Wilson, 2014; Ruehl et al., 2016) and field measurements (Ovadnevaite et al., 2017), and applied to SSA analogues (Forestieri et al., 2018). The compressed film model creates an organically enriched surface layer, which acts to suppress surface tension, and a bulk solution which is a mixture of organic species, inorganic species and water. With a sufficient concentration of a surface active species the water tension suppression can compensate for the reduction in water

uptake due to the Raoult effect. As the droplet grows the organic monolayer becomes increasingly dilute, until the droplet surface tensions approaches the surface tension of water. The application of the compressed film model for SSA analogues indicated that the organic component can be considered dissolved into the bulk (Forestieri et al., 2018; Fuentes et al., 2011; Petters and Petters, 2016; Prisle et al., 2010). The surface tension suppression of SSA is far from consistent suggesting that regional differences and the richness in organic matter and/or microbial composition are important.

Measurements of nascent SSA composition and water uptake taken during the Surface Ocean Aerosol Production (SOAP) research voyage (Law et al., 2017) over the Chatham Rise in the South-West Pacific Ocean, on-board the RV Tangaroa (NIWA, Wellington), are reported here. The measured SSA composition and hygroscopicity are presented with respect to an extensive suite of seawater composition. Aitken mode SSA composition was inferred from volatility data and $PM_1$ composition was measured from analysis of filter samples. These data provide a valuable comparison between Southern Hemisphere observa-

tions and existing models for SSA organic enrichment and SSA water uptake.



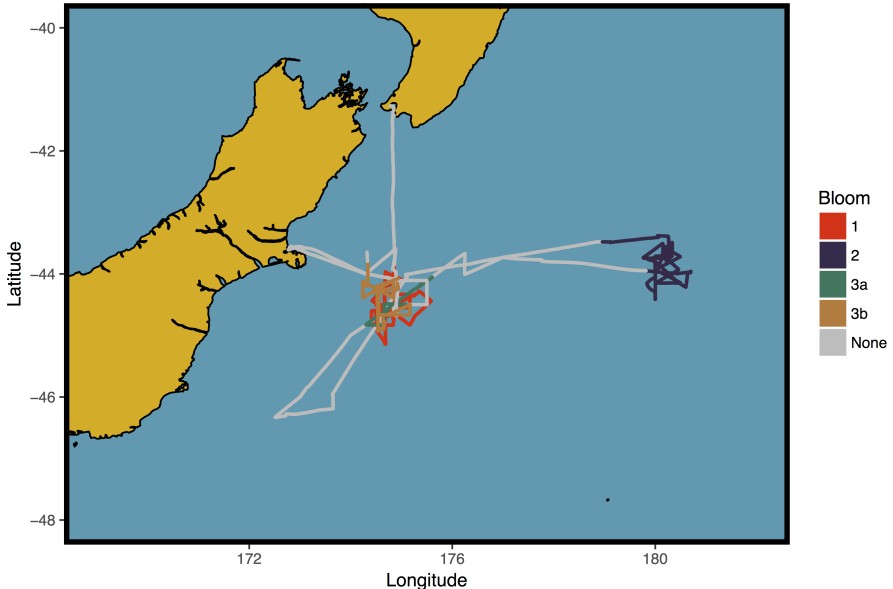

**Figure 1.** Voyage map for SOAP study, coloured by bloom periods.

## 2 Methods

### 2.1 SOAP voyage

The SOAP voyage examined air-sea interactions over the biologically-productive frontal waters of the Chatham Rise, east of New Zealand in February and March 2012. The Chatham Rise couples pristine marine air masses with high biological activity due to the mixing of warm subtropical water and cool Southern Ocean waters. Subtropical waters have relatively low macronutrient levels, while Southern Ocean waters are depleted in iron, but not in macronutrients (Law et al., 2017). Phytoplankton blooms were identified via satellite ocean colour images and further mapped using continuous measurement of seawater parameters (Chl $a$, $\beta_{660}$ backscatter, $pCO_2$, $DMS_{sw}$). Three broad bloom periods were defined as shown in Fig. 1. The first bloom (B1) occurred 12 hours in to the voyage and was characterised by dinoflagellates and displayed elevated Chl $a$ and seawater DMS, 7 days into the voyage a weakening bloom (B2) was driven by coccolithophores, and a final bloom (B3) displayed a mixture of phytoplankton groups. Bloom 3 was subdivided into B3a and B3b due to changes in the surface water characteristics following the passage of a storm.

### 2.2 Measurements/instrumentation

Seawater samples were collected throughout the voyage for the purpose of generating nascent SSA. Seawater was primarily collected from the ocean surface (approximately 10 cm depth) during workboat operations at a distance from the RV Tangaroa or from the mixed layer (3 - 12 m depth, always less than the measured mixed layer depth) from a conductivity, temperature




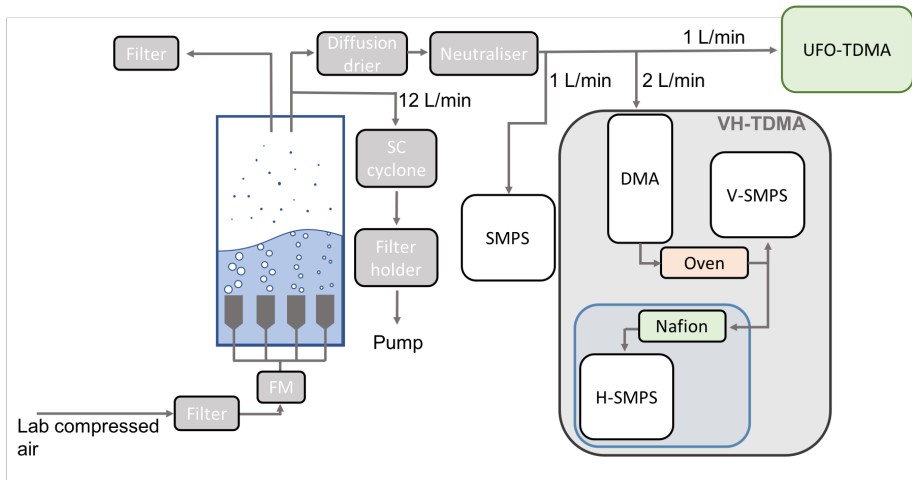

**Figure 2.** Experimental schematic of nascent SSA chamber experiments during SOAP voyage. The VH-TDMA (grey) contains an RH controlled region (blue) used for water uptake measurements.

and depth rosette (CTD), and a number of deep water samples were collected for comparison. Due to the sampling method used these seawater samples are not representative of the sea surface microlayer (SML). Samples also spanned the variability in ocean biological conditions observed throughout the the SOAP voyage. Table S1 provides a description of the seawater samples that were taken.

Nascent SSA was generated in-situ in a 0.45 m$^3$ cylindrical polytetrafluoroethylene chamber housing four sintered glass filters with porosities between 16 and 250 $\mu m$ (Cravigan, 2015; Mallet et al., 2016). Dried and filtered compressed air was passed through the glass filters at a flow rate of $15.5 \pm 3$ L min$^{-1}$ and resulting SSA was sampled from the headspace of the chamber. A diffusion drier was used to dry the sample flow to $20 \pm 5$ % RH prior to characterisation. Figure 2 shows the sampling set up used to generate, condition and measure nascent SSA.

Size distributions and number concentrations of 10 to 300 nm diameter SSA were measured using a TSI 3080 scanning mobility particle sizer (SMPS), coupled with a 3071 differential mobility analyser and a 3010 condensation particle counter (CPC) (TSI, Shoreview, MN), with an aerosol sample flow rate of 1 L min$^{-1}$ and a sheath flow rate of 10 L min$^{-1}$. The volatility and hygroscopicity of nascent SSA was determined with a volatility and hygroscopicity tandem differential mobility analyser (VH-TDMA) (Alroe et al., 2018; Johnson et al., 2008, 2004). The VH-TDMA selects particles based on mobility

diameter, conditions them, and measures the resulting particle size distributions using parallel SMPSs (V-SMPS and H-SMPS in Fig. 2), each with a TSI 3010 CPC. The aerosol sample flow rate for each SMPS was 1 L min$^{-1}$, resulting in a total inlet flow of 2 L min$^{-1}$, the sheath flow for the pre-DMA, V-DMA and H-DMA were 11, 6 and 6 L min$^{-1}$, respectively. The VH-TDMA can be operated in a number of sampling modes, although in general the instrument is designed to observe the water uptake at ambient temperature and subsequently observe the water uptake of a non-volatile component (at some elevated

temperature), which is used to infer the water uptake of the volatile component. Nascent SSA with a mobility diameter of 50



nm were preselected for each water sample and a number of water samples were also analysed with a preselected diameter of 30, 100 or 130 nm for comparison. Table S1 details the VH-TDMA analysis pre-selected particle size. The SSA volatile fraction was determined by measuring the diameter of preselected SSA upon heating by a thermodenuder up to 500 °C, in temperature increments of 5 °C - 50 °C. Subsequent to heating the SSA was exposed to 90% RH and the hygroscopic growth

factor was measured. The dependence of HGF on RH at ambient temperature was measured for one water sample (workboat 9) to provide the deliquescence relative humidity (DRH).

Size distributions (Fig. 4), volatility profiles (Fig. S2) and HGFs have also been measured for laboratory sea salt samples, which were generated using the same glass filters, chamber and sample conditioning. In addition laboratory sea salt Transmission Electron Microscopy (TEM) samples were collected using a TSI 3089 Nanometer Aerosol Sampler and analysed using

X-ray dispersive spectrometry (TEM-EDX). TEM data were collected on a JEOL2100 transmission electron microscope operating at 200 kV coupled with a Gatan high angle annular dark field (HAADF) detector. TEM images were used to compare the morphology of heated (250 °C) and unheated SSA samples and TEM-EDX data were used to compute enrichment factors for laboratory sea salt.

The ultrafine organic tandem differential mobility analyzer (UFO-TDMA, Vaattovaara et al., 2005) was used to calculate

moderately oxidized organic volume fractions of SSA particles by measuring how much the particle grows in sub-saturated (82% ± 2%) ethanol vapour. The growth factor of the SSA samples in ethanol vapour was measured at pre-selected mobility diameters of 15, 30, 40 and 50 nm using the UFO-TDMA. UFO-TDMA and H-TDMA measurements of sample U7520 were pre-treated using a thermodenuder (heated in 10°C steps up to 500°C). This allowed examination of the contribution of volatile components to particle growth in ethanol vapour, but excluded the estimation of organic fractions from VH-TDMA

measurements.

SSA generated from 23 ocean water samples was collected on filters for further compositional analysis using transmission Fourier Transform Infra Red (FTIR) and Ion Beam analysis (IBA). SSA was sampled through a 1 $\mu m$ sharp cut cyclone (SCC 2.229PM1, BGI Inc., Waltham, Massachusetts) and collected on Teflon filters, with the sample confined to deposit on a 10 mm circular area. Back filter blanks were used to characterise the contamination during handling, and before analysis samples

were dehydrated to remove all water, including SSA hydrates, as described in Frossard and Russell (2012). Filter blanks were under the detection limit for the FTIR and Si was the only compound with blank measurements above the IBA detection limit. FTIR measurements were carried out according to previous marine sampling techniques (Maria et al., 2003; Russell et al., 2011, 2010) and characterised the functional groups associated with major carbon bond types, including saturated aliphatic (alkane) groups, alcohol (used here to include phenol and polyol) groups, carboxylic acid groups, non-acidic carbonyl groups,

and primary amine groups. FTIR measurements are non-destructive, therefore subsequent to FTIR analysis filter samples underwent simultaneous particle induced X-ray emission (PIXE) and gamma ray emission (PIGE) analysis (Cohen et al., 2004). The elements discussed herein, of interest for SSA, are Na (from PIGE) and Mg, Si, S, Cl, K, Ca, Zn, Br and Sr (from PIXE). It should be noted that Rutherford backscattering and particle elastic scattering analysis did not yield useful results for the analysis of C, N, O, and H. The SSA organic concentrations were instead obtained solely from FTIR analysis.





A large number of ocean water measurements were taken, characterising the physical properties, nutrient concentration, the phytoplankton population, bloom productivity and the concentration of molecular classes important for SSA e.g. fatty acids, proteins and carbohydrates. A detailed list of ocean water measurements undertaken during the SOAP voyage is contained in Law et al. (2017). The parameters of interest here are the concentrations of Chl $a$, DOC, high molecular weight proteins and

sugars, alkanes and the fatty acid concentration. It should be noted that the protein and carbohydrate measurements include both dissolved and particle components. Fatty acid measurements are made up of measurements from 34 individual fatty acid species, which can be broken up into saturated (14 species), monounsaturated (9 species) and polyunstaurated (11 species). Alkanes were also speciated, with carbon numbers ranging from 13 to 36. Details of the sampling and analysis methodology for the parameters used in this paper are available in the supplementary material.

## 10  2.3  Data analysis

Nascent SSA size distributions for each water sample were averaged and normalised to their maximum value. Non-linear least square fits of up to four lognormal modes were fitted to each distribution with a random selection of initial values for the geometric mean and standard deviation, constrained to 10 - 320 nm and less than 2, respectively. The most appropriate fit was determined using the bayesian information criterion, which is a measure of the error in reconstructing the measured size

distribution that applies a penalty based on the number of parameters used and therefore avoids over fitting (Sakamoto et al., 1987).

All VH-TDMA data were inverted using the TDMAinv algorithm (Gysel et al., 2009), and external modes were allocated based on local maxima of the resulting piecewise linear GF distribution. The volatile fraction (VF) was computed using Equation 1, where d is the particle diameter, T denotes the temperature of the thermodenuder and o denotes ambient temperature.

$$VF = 1 - VFR = 1 - \left(\frac{d_T}{d_0}\right)^3 \tag{1}$$

In this study SSA organic volume fractions were calculated using volatility measurements, by accounting for the presence of sea salt hydrates. The volatility due to hydrates was used as a proxy for the proportion of inorganic sea salt in the natural seawater samples, which in turn provided the proportion of organics. As the organic fraction of internally mixed SSA increases, the sea salt fraction decreases and the hydrate fraction decreases in proportion to the sea salt. The sea salt fraction was computed

by comparing natural sea spray volatility profiles and laboratory sea salt volatility profiles. In this study it was assumed that volatility was due to the evaporation of hydrates over the temperature range 200 - 400 °C, i.e. that semi-volatile organics evaporate at temperatures less than 200 °C and non-volatile organics evaporate at temperatures above 400 °C. This assumption is supported by previously reported volatility profiles of nascent SSA generated from natural seawater and from laboratory sea salts, which are consistently parallel at 200 - 400 °C, indicating that the volatility is due to the evaporation of similar

components i.e. hydrates (Modini et al., 2010b; Rasmussen et al., 2017).

The organic volume fraction was inferred from volatility measurements using the linear model outlined in Eq. 2, and shown in Fig. S2, where $VF_T$ is the measured volatile fraction of the sea spray sample at thermodenuder temperature T and $VF_{T,SS}$ is



the measured volatile fraction of laboratory sea salt at temperature T. The slope of Eq. 2, $f$, is the proportion of volatility due to sea salt hydrates, and was assumed to represent the proportion of hydrated sea salt in the natural seawater samples. The total organic volume fraction is then given by $1 - f$. For example an internally mixed SSA particle with an organic volume fraction of 50% and a sea salt (including hydrate) volume fraction of 50% has half the volume of hydrates compared the laboratory sea

salt particles of the same diameter. The volatility due to hydrates will be reduced by half and therefore $f$ will also be reduced by half relative to laboratory sea salt, assuming that hydrates dominate the volatility at 200 - 400 °C. Further detail on the linear fits for individual samples is shown in the supplementary material.

$$VF_{T=200-400^\circ C} = f \times VF_{T=200-400^\circ C,SS} + OVF_{sv} \qquad (2)$$

Laboratory sea salt volatility profiles were measured using three different sea salt samples, a commercially available sea salt (Pro Reef Sea Salt, Tropic Marin, Wartenberg, Germany), and two mixtures of laboratory grade salts, one mimicking Sigma-Aldrich Sea Salts composition and one mimicking the Niedermeier et al. (2008) Atlantic Ocean sea salt composition. Sea salt solutions were all made to a concentration similar to sea water, $35\,\mathrm{g\,L^{-1}}$ and volatility profiles were within experimental error of one another. The error in the laboratory sea salt volatile fractions were assumed to be the maximum of the standard error in

the mean across the three sea salt samples and the instrumental error ($\pm\,3\%$).

The method used to compute the organic volume fraction implicitly assumes that the proportion of hydrates in the sea salt component of SSA is constant, however observations have shown variability in inorganic composition of SSA can vary (Salter et al., 2016; Schwier et al., 2017; Ault et al., 2013a), particularly species such as Ca, Cl and Mg, which are potentially important for the formation of hydrates. As a result a further correction was applied to $f$ for the case where the composition of

inorganic sea salt, and therefore hydrates, is different between the natural SSA sample and the laboratory sea salt sample. The correction is represented by $f_{io}$ in Eq. 3. Ca and Mg enrichment and associated Cl depletion can result in an overall reduction in hydrate forming sea salt species, such as $CaCl_2$ and $MgCl_2$ (Salter et al., 2016; Schwier et al., 2017). The ionic composition of nascent SSA generated from natural seawater was measured using ion beam analysis and used to compute the inorganic molecular composition, which in turn was used to compute the volume fraction of hydrates for each sample as described in the

supplementary material. The same analysis was performed based on the ionic composition of laboratory sea salts and $f_{io}$ was computed as the ratio of natural seawater SSA hydrate volume fraction to laboratory sea salt hydrate volume fraction.

$$OVF_{tot} = 1 - \frac{f}{f_{io}} \qquad (3)$$

The volume fraction of semi-volatile organics in the nascent SSA generated from natural seawater, which evaporate at temperatures less than 200 °C, were computed from the intercept in Eq. 2. Uncertainties for $OVF_{sv}$ and $OVF_{tot}$ were taken

to be whichever is the maximum out of the measurement error for VF and the standard error for the intercept and slope, respectively. The measurement error in VF is $\pm 3\%$, which is due to a 1% DMA sizing uncertainty (Johnson et al., 2004;





Modini et al., 2010a). This approach quantifies the proportion of volatility due to the presence of hydrates and due to the presence of organics, and is therefore an improvement on previously published estimates of the SSA OVF (Mallet et al., 2016; Modini et al., 2010a; Quinn et al., 2014).

The HGF was computed using Eq. 4, where $d_{RH,T}$ is the measured diameter at the RH of the H-SMPS (90% for all
except the DRH measurement) and thermodenuder temperature T, and $d_{dry,T}$ is the measured dry diameter at thermodenuder temperature T. HGFs for sea salt were shape corrected using the dynamic shape factor from (Zieger et al., 2017). The presence of an organic fraction has been observed to increase the sphericity of nascent SSA (Laskin et al., 2012), therefore an organic fraction dependent shape correction was applied (Zelenyuk et al., 2007). A single shape factor was used across all temperatures because TEM images of laboratory sea salt showed an insignificant difference between the apparent shape of SSA at ambient
temperature and those heated to 250 °C (Figure S4).

$$HGF = \frac{d_{RH,T}}{d_{dry,T}} \qquad (4)$$

The organic mass fraction from SSA samples collected on filters was computed from the total organic mass from FTIR analysis and the inorganic mass from ion beam analysis, as in Eq. 5. The filter exposed area (0.785 $cm^2$ was used to convert inorganic areal concentrations into total mass. The inorganic mass (IM) was computed as the sum of Na, Mg, $SO_4$, Cl, K, Ca,
Zn, Br, Sr, and the $SO_4$ mass was computed by multiplying the S mass by 3 i.e. all S was assumed to be in the form of $SO_4$. The uncertainty in the organic mass measured using FTIR is up to 20% (Russell, 2003; Russell et al., 2010), this is taken as the uncertainty in OMF.

$$OMF = \frac{OM_{FTIR}}{OM_{FTIR} + IM_{IBA}} \qquad (5)$$

Enrichment factors for inorganic elements were calculated with respect to the laboratory prepared seawater and presented
with respect to $Na^+$(Salter et al., 2016), as shown in Eq. 6, where X is the element of interest. Enrichment factors were calculated from Ion Beam Analysis of filter samples and from TEM-EDX analysis of laboratory sea salt samples.

$$EF(X) = \frac{([X]/[Na^+])_{SSA}}{([X]/[Na^+])_{water}} \qquad (6)$$

The OCEANFILMS model was implemented for the surface and mixed layer nascent SSA experiments with measured water parameters used to represent bulk seawater molecular classes. Lipids were assumed to be equal to the total concentration of fatty
acids, the total high molecular weight proteins were used to represent the protein molecular class, and high molecular weight reducing sugars were used to represent polysaccharides. Note that these measurements were not micro layer measurements, the seawater samples were collected via CTDs or on workboats. Missing water composition data were filled using the relationships outlined in Burrows et al. (2014), based on the lifetime of each molecular class, for example the bulk concentration of proteins was assumed to be equal to one third of the polysaccharide concentration, when no other data were available. The humic like





molecular class was assumed not to be present at the surface and in the mixed layer, and therefore not included (Burrows et al., 2014). The processed molecular class concentration was assumed to make up the remainder of the dissolved organic carbon (DOC) after polysaccharides, proteins and lipids have been subtracted, a minimum was applied to the concentration of processed compounds to prevent unrealistically low or negative concentrations. The Langmuir adsorption coefficients for

each molecular class was taken directly from Burrows et al. (2014). OCEANFILMS was run including the co-adsorption of polysaccharides with all other molecular classes (Burrows et al., 2016), and an assumed bubble thickness of 0.3 $\mu m$. The effect of the bubble thickness is to change the ratio of organics to salt in the bubble film and therefore the SSA organic fraction, however the distribution of organic molecular classes don't vary. The assumed bubble thickness was not based on any measured bubble parameters.

The ZSR approximation was used to compute the ambient temperature nascent SSA HGF, defined as the average of all measurements less than 45°C, and the heated nascent SSA HGF, defined as the average HGF for all measurements between 255 - 405 °C. The terms used in the ZSR mixture were sea salt, non-volatile organics, semi-volatile organics and hydrates. The volume fraction of each of these components was based on the volatility measurements (Eq. 2 and Eq. 3), the hydrate fraction was computed from the difference between the VF at 255 - 405 °C and the $OVF_{sv}$. The HGFs for the hydrate component,

semi-volatile organic and non-volatile organic component were assumed to be 1 (Modini et al., 2010a) and the HGF for the sea salt component was assumed to be 2.15 at 50 nm, based on measurements of heated laboratory sea salt 2.15 ± 0.06 averaged across 300 - 350 °C. The sea salt HGF is also consistent with the HGF of pure NaCl (Zieger et al., 2017), which represents the HGF of sea salt without hydrates. HGFs have not been kelvin corrected here because all measurements were performed at the same pre-selected particle size. The hydrate fraction of the salt has been explicitly included here to account for any

variation in the proportion of hydrates between samples. Calculations using the ZSR approach were repeated using an organic (semi-volatile and non-volatile) HGF of 1.6 as a sensitivity test for the relationship between OVF and HGF. The HGF of 1.6 was chosen as an upper limit for organic-salt mixtures that could possibly be present in SSA (Estillore et al., 2016, 2017).

As a counterpoint to the ZSR assumption, which assumes the organic component is dissolved into the bulk, the compressed film model (Ruehl et al., 2016) was applied to explore the influence of partitioning organics to the surface on the nascent SSA

water uptake. The composition of the SSA organics are unknown, therefore the compressed film model was computed for organics with molecular areas ($A_0$) ranging from 10 - 200 square angstroms and molecular volumes ($V_{org}$), ranging from 0.2 to 3.6 $\mathrm{m^3\,mol^{-1}}$. The speciation of organics into molecular classes was calculated from the functional group concentrations as shown in Burrows et al. (2014). HGFs were computed for three cases, assuming that just the lipids partition to the surface, that the lipids and the polysaccharides partition to the surface and assuming that all of the organics partition to the surface. The

hygroscpicity of the bulk aerosol i.e. the component not partitioned to the surface, was computed using the ZSR assumption as outlined in the previous paragraph, with an organic HGF of 1.

The output from the compressed film model, with all of the organics partitioned to the surface, was used to calculate the CCN concentration, and was compared to the CCN concentration computed from the ambient HGFs, assuming that the droplet surface tension is equal to the surface tension of water. The critical supersaturation from the compressed film model, with all

of the organics partitioned to the surface, was used to compute the critical diameter using the $\kappa$-Köhler equation (Petters and



Kreidenweis, 2007). The CCN number concentrations were subsequently estimated assuming a nascent SSA size distribution with number concentration $100\,\mathrm{cm^{-3}}$, mean diameter of 160 nm and geometric standard deviation of 2.6. CCN concentrations were computed by integrating the size distribution at diameters greater than the critical diameter, which was computed (as above) for the compressed film model and assumed to be equal to the preselected particle diameter for the full solubility

(surface tension equal to surface tension of water) case.

## 3   Results

### 3.1   Seawater composition

Chl $a$ concentrations from water samples used to generate SSA ranged from 0.29 to $1.53\,\mathrm{\mu g\,L^{-1}}$ as shown in Fig. 3, which are indicative of productive open ocean regions (O'Dowd et al., 2015), in particular for the Southern Hemisphere. Measured Chl $a$

concentrations are up to an order of magnitude lower than previous SSA measurements taken in coastal waters (Frossard et al., 2014; Quinn et al., 2014). Significant correlations were observed between Chl $a$ and total high molecular weight proteins and polyunsaturated fatty acids ($R^2 = 0.51$, p-value $< 0.01$). The saturated fatty acid component was the largest contributor to the total fatty acid concentration, and was made up of stearic, palmitic, myristic and lauric acid ($C_{18}$ to $C_{12}$, all even). Monounsaturated fatty acids were dominated by oleic acid ($C_{18}$) and polyunsaturated fatty acids were made up of docosahexaenoic and

eicosapentenoic acid ($C_{22}$ and $C_{20}$). Fatty acid concentrations showed significant correlation with the concentration of alkanes ($R^2 = 0.75$, p-value $< 0.001$), particularly monounsaturated fatty acids. Alkanes displayed even carbon numbers from 16 to 28, with peak concentrations for octadecane ($C_{18}$) and eicosane ($C_{20}$). It is worth noting that fatty acid and alkane measurements were only done for the surface and mixed layer samples and were not done for the deep water samples.

     Bloom 1 was dominated by dinoflagellates and displayed the highest Chl $a$ concentrations of $0.84 \pm 0.2\,\mathrm{g\,L^{-1}}$ and total

phytoplankton carbon concentrations. Relatively short lived aliphatic species, such as fatty acids and alkanes were elevated during bloom 1, as were proteins, which are of intermediate lifetime. High molecular weight reducing sugars were elevated during bloom 1, although less so than the proteins. Elevated surface concentrations were also noticeable during bloom 1, particularly for the aliphatic species, in particular alkanes with 3.1 times higher average concentration from surface measurements ($\sim$0.1m) than from shallow mixed layer measurements ($\sim$2m), and fatty acids with 1.7 (saturated) to 4.2 (monounsaturated)

times higher surface concentrations. Elevated surface concentrations were also observed for Chl $a$ (1.7 times higher) and carbohydrates (1.5 times higher). The apparent gradient in organics in the surface seawater is distinct for bloom 1, and points to to a potentially enhanced contribution from surface active species over this bloom.

     Bloom 2 was characterised as a coccolithophore bloom (Law et al., 2017) and displayed decreasing Chl $a$, fatty acid and high molecular weight protein concentrations throughout the bloom. The overall organic concentrations in bloom 2 were lower

than bloom 1, with average Chl $a$ concentrations of $0.67 \pm 0.3\,\mathrm{g\,L^{-1}}$, however DOC concentrations were slightly elevated ($807 \pm 65\,\mathrm{g\,L^{-1}}$) compared to bloom 1 ($714 \pm 135\,\mathrm{g\,L^{-1}}$) and bloom 3 ($718 \pm 84\,\mathrm{g\,L^{-1}}$). The number of measurements during bloom 3 were limited, however initially the bloom displayed similar concentrations to bloom 2, Chl $a$ of $0.44 \pm 0.17\,\mathrm{g\,L^{-1}}$





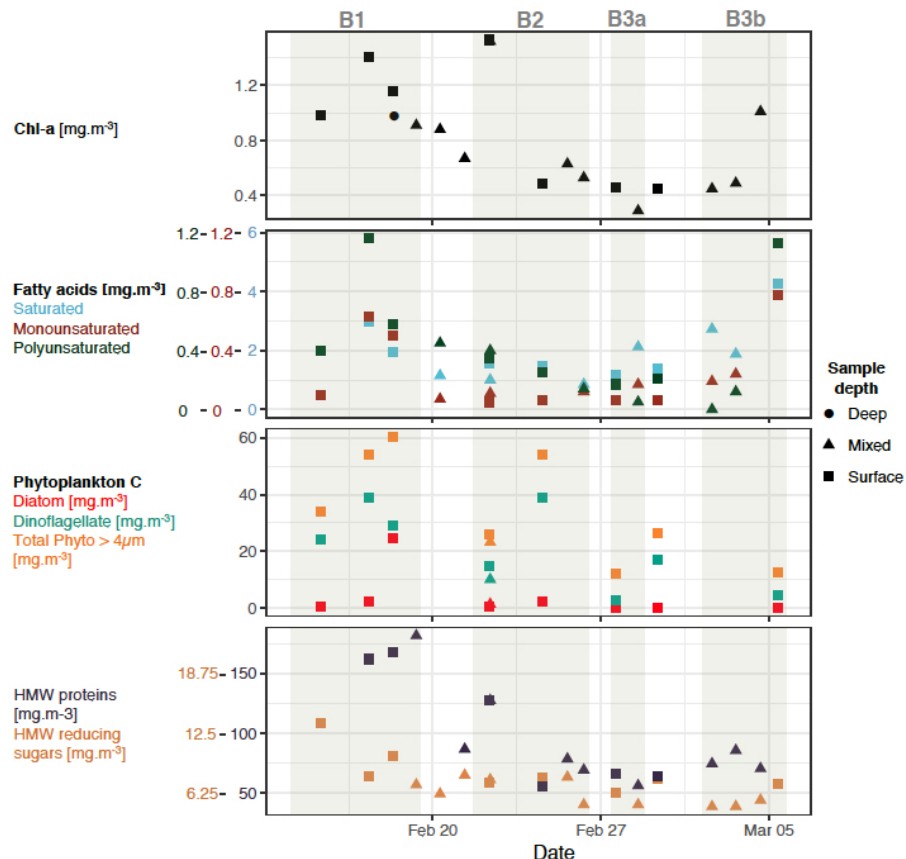

**Figure 3.** Characterisation of biological activity for water samples used to generate SSA. Note that this is a selected subset of all water parameters. Panels show the concentration of Chl $a$ (top), fatty acids (2nd from top), phytoplankton carbon (2nd from bottom) and total high molecular weight (HMW) reducing sugars and proteins (bottom). Note that total HMW proteins and reducing sugars include particle and dissolved fractions. Shapes represent the water sample depth class (surface 0.1 m, mixed 3 - 12 m and deep > 12 m).

and the final measurements of bloom 3b displayed elevated Chl $a$ and fatty acid concentrations. Bloom 3 displayed the lowest total phytoplankton carbon concentrations.

### 3.2 SSA size distributions

The measured size distributions were broken up into four log-normal modes characterised by geometric mean diameters ranging from 33 to 320 nm, as seen in Fig. 4. Natural SSA size distributions were slightly shifted towards larger diameters compared to laboratory sea salt measurements, and showed a significant enhancement in mode 2 and lower contributions from modes 1 and 4 (Fig. 4 and Table 1). The size distribution of SSA generated from natural seawater samples is more narrow than laboratory sea salt particle size distributions, which is consistent with the addition of a surfactant material (Fuentes et al., 2010; Modini





**Table 1.** Nascent SSA lognormal parameters

| Water sample | Parameter | Mode 1 | Mode 2 | Mode 3 | Mode 4 |
|---|---|---|---|---|---|
| Laboratory sea salt | Normalised number conc. | 0.34 | 0.12 | 0.40 | 0.14 |
| | Mean diameter | 40 | 69 | 114 | 309 |
| | Geometric standard deviation | 1.61 | 1.29 | 1.63 | 1.31 |
| Natural seawater (average) | Normalised number conc. | 0.18 | 0.38 | 0.38 | 0.05 |
| | Mean diameter | 34 | 70 | 120 | 320 |
| | Geometric standard deviation | 1.47 | 1.45 | 1.58 | 1.44 |

et al., 2013) which allows the saline components of the bubble film to drain more before bursting, producing an organically enriched particle with a more uniform distribution.

The shape of the nascent SSA size distribution was broadly similar to nascent SSA size distributions observed in previous studies, but shifted to slightly larger mean diameters. For example Fuentes et al. (2010) fitted lognormal modes with mean

mobility diameters of 20, 41, 87 and 250 nm to laboratory sea salt generated using glass sintered filters, and modes with mean mobility diameters of 14, 48, 124 and 334 nm for plunging water generated sea salt. SSA produced from sintered glass filters does not perfectly represent real world bubble bursting from wave breaking (Collins et al., 2014; Prather et al., 2013) but the use of four glass filters with different pore sizes resulted in a broader distribution than other measurements of nascent SSA using glass filters (Collins et al., 2014; Fuentes et al., 2010; Keene et al., 2007; Mallet et al., 2016). Observations have shown

organic enrichment (King et al., 2013) and also externally mixed organics (Collins et al., 2014) for Aitken and accumulation mode SSA using sintered glass techniques, with slightly higher organic enrichment than that observed using plunging water or wave breaking methods. Despite the limitations, the use of sintered glass filters allowed an examination of the components of seawater that contribute to SSA organic enrichment.

### 3.3   SSA composition

Volatility measurements using the VH-TDMA indicated that the SSA volatile organic fraction made up a relatively consistent proportion of the 50 nm SSA, with a $OVF_{SV}$ of $0.11 \pm 0.04$ (mean $\pm$ sd), as shown in Fig. 5. SSA compositional results are also tabulated in the supplementary material. The non-volatile organic fraction, however, made up a much larger and more variable proportion, with an average $OVF_{NV}$ of $0.39 \pm 0.24$. The 50 nm OVF was highest during bloom 1 (generally greater than 0.6), which is coincident with seawater samples enriched in organics, during which time non-volatile organics dominated.

A dominant non-volatile organic SSA fraction has been observed for nascent SSA measurements in the North Pacific and North Atlantic oceans (less than 15% volatilised at below 230 °C), and in the Great Barrier Reef (Bates et al., 2012; Mallet et al., 2016; Quinn et al., 2014), our observations are broadly consistent with these results. It should be noted that a non-volatile SSA fraction is not universally observed (Modini et al., 2010a; O'Dowd et al., 2004; Ovadnevaite et al., 2011b). Organic mass fractions measured using FTIR and ion beam analysis of filter samples averaged $0.12 \pm 0.6$, and were elevated during bloom





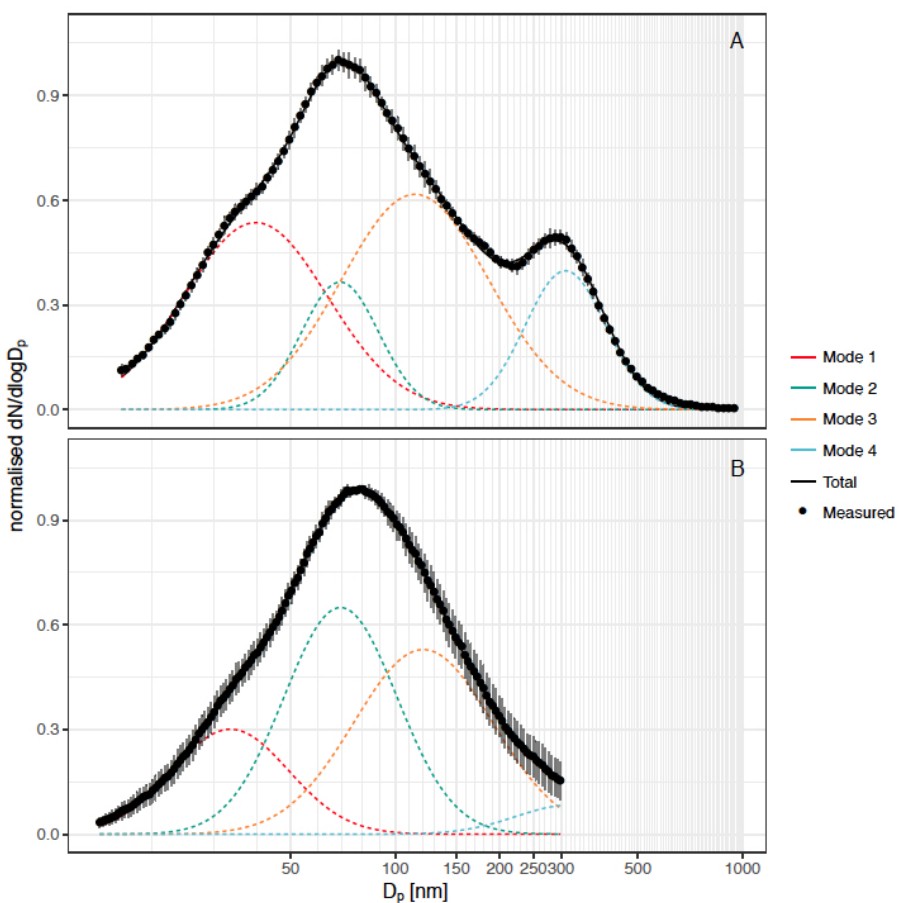

**Figure 4.** Nascent SSA size distributions from laboratory sea salt measurements (A) and natural seawater measurements (B). Seawater size distributions are an average of all water samples. Lines represent the fitted modes (dotted) and total fitted distribution (solid), black dots represent measured values and grey bars indicate the standard deviation in dN/dlogDp.

1 with surface water samples displaying OMF of approximately 0.2. The values observed here fit within the broad range of observed OMFs for nascent SSA, for example at Mace Head summertime SSA organic mass fractions of up to 0.8 have been observed (Facchini et al., 2008; O'Dowd et al., 2004), while summertime OMFs of 3 - 7% have been observed for the North Atlantic and North Pacific Oceans (Bates et al., 2012; Quinn et al., 2014).

5    The OVFs measured using volatility were applied to the lognormal distributions shown in Fig. 4 assuming that the first two lognormal modes contained a volatile and a non-volatile organic volume fraction, lognormal mode 3 contained a volatile organic fraction only and mode 4 was composed of inorganic sea salt. These compositions were applied based on the mixing state from HGF measurements (see Section 3.4 *SSA water uptake*), which indicated an external mixture. Note that the external mixture wasn't observed in the volatility measurements, this is likely to be due to the relatively small observed volatile fraction

10   and the resulting limitation on the sensitivity of volatility measurement. The organic density was assumed to be $1.1\,\mathrm{g\,cm^{-3}}$

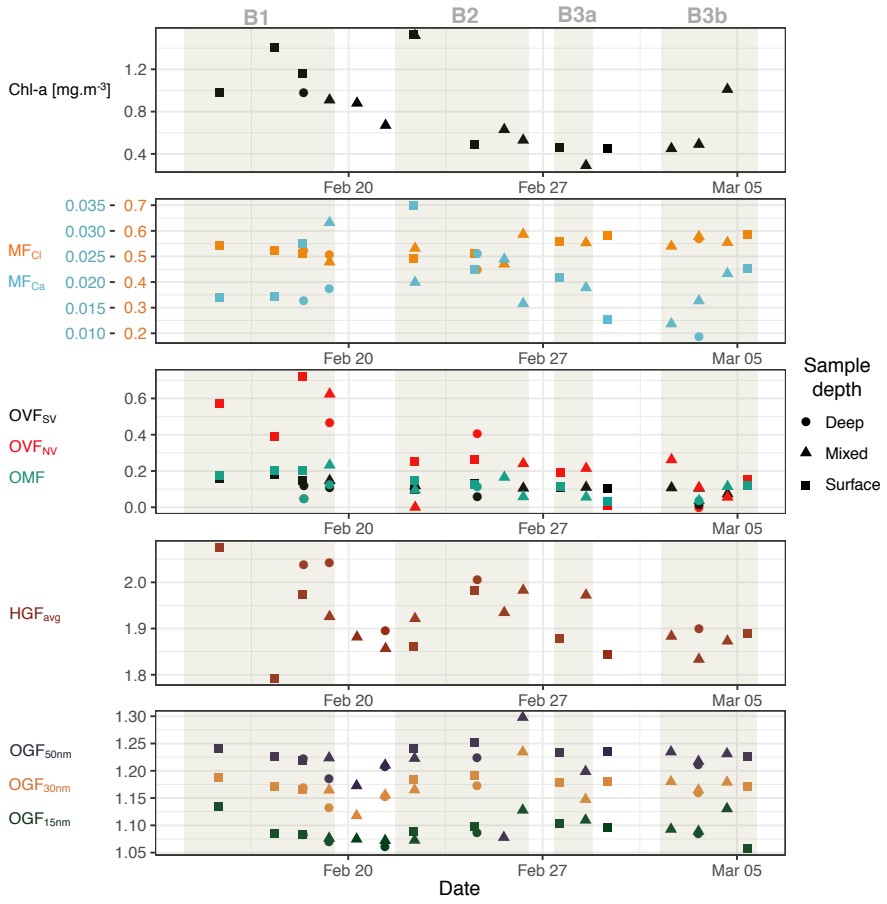

**Figure 5.** Summary of nascent SSA properties. Chl $a$ time series included for context on phytoplankton bloom conditions (top panel). Mass fractions (MF) of $Cl^-$ and $Ca^{2+}$ (multiplied by 20) relative to total inorganic mass concentrations measured using IBA on filter samples (2nd panel from top). VH-TDMA $OVF_{SV}$, $OVF_{NV}$ and FTIR/IBA derived OMF from filter samples (3rd panel from top). HGF measured using VH-TDMA (2nd panel from bottom), organic growth factor measured using UFO-TDMA (bottom panel).

(Keene et al., 2007; Modini et al., 2010a) and the sea salt density was assumed to be $2.01 \; g \, cm^{-3}$ (Zieger et al., 2017). Figure 6 shows that the organic fraction inferred using volatility techniques and the fraction measured using FTIR/IBA of the filter samples agree reasonably well. The organic mass fraction calculated using the aerosol volatility and size distributions overestimates the organic mass fraction (Fig. 6) by a factor of 1.3 on average. There are a number of sources of uncertainty, including the assumed partitioning to lognormal modes and the particle densities. The presence of inorganic species, such as $Ca^{2+}$, complexed with the organics are included in organic estimates from volatility, but not from filter analyses.

Correlations of both the semi volatile organic volume fraction ($OVF_{SV}$) and the non-volatile organic volume fraction ($OVF_{NV}$) with seawater high molecular weight proteins were observed ($R^2$ of 0.43 and 0.44, p-value $< 0.01$, and slope of $3 \pm 1 \times 10^{-3}$ and $4 \pm 1 \times 10^{-4}$, respectively). Similarly for high molecular weight carbohydrates ($OVF_{SV}$ $R^2$ of 0.4, p-value $<$



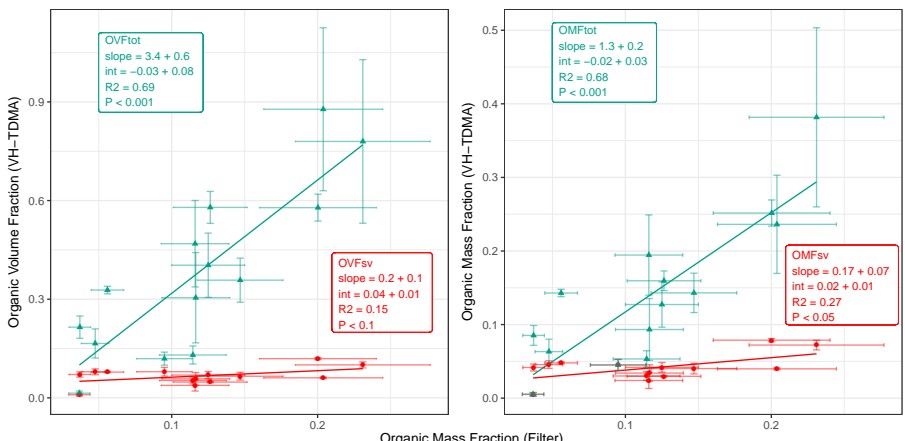

**Figure 6.** Comparison of 50 nm organic volume fraction (left) and PM$_1$ organic mass fraction (right) calculated from volatility measurements (using VH-TDMA) and the OMF measured using FTIR/IBA on filter samples. Volatile fractions shown in red, total organic fractions shown in green. VH-TDMA derived PM$_1$ organic mass fractions were calculated from the organic volume fraction measurements using size resolved composition assumptions discussed in Section 3.3.

0.01, and slope of $8 \pm 2 \times 10^{-3}$, OVF$_{NV}$ R$^2$ of 0.28, p-value $< 0.05$, and slope of $5 \pm 2 \times 10^{-2}$). In addition the semi volatile OVF correlated with total alkanes (R$^2$ of 0.46, p-value $< 0.01$, and slope of $8 \pm 2 \times 10^{-3}$) and polyunsaturated fatty acids (R$^2$ of 0.39, p-value $< 0.05$, and slope of $4 \pm 2 \times 10^{-2}$). Correlations with water parameters suggest that the composition of the volatile and non-volatile OVFs were similar, but the semi-volatile OVF displayed a higher contribution from aliphatic, lipid

like species. The correlation between semi-volatile OVF and seawater alkanes was significant for all carbon numbers between C$_{16}$ and C$_{26}$. The strongest correlations were observed for lower carbon numbers, for C$_{16}$ the R$^2$ was 0.55, the p-value was $<$ 0.001, and the slope was $0.07 \pm 0.02$, and for C$_{26}$ the R$^2$ was 0.28, the p-value was $< 0.05$, and the slope was $0.17 \pm 0.07$. Concentrations were largely below the detection limit for carbon number greater than C$_{26}$.

The mass fraction of inorganic species in SSA during SOAP was observed to vary from that of salts in seawater, in particular

an enrichment factor of $1.7 \pm 0.6$ relative to the composition of laboratory sea water was observed for Ca$^{2+}$ and $0.4 \pm 0.2$ for Mg$^{2+}$ (Fig. 7). Enrichment factors (EF) observed from TEM-EDS analysis of laboratory sea salt samples were $0.8 \pm 0.3$ for Ca$^{2+}$ and $1.0 \pm 0.1$ for Mg$^{2+}$ ($mean \pm sd$), suggesting sea salt fractions similar to seawater. It should be noted that TEM-EDS EF were based on a modest number of measurements (25 particles). Ca$^{2+}$ inorganic mass fraction and EF were observed to increase with OMF, while the Cl$^-$ inorganic mass fraction and EF decreased with increasing OMF. The mass ratio of Cl$^-$

to Na$^+$ was $1.6 \pm 0.2$, which is slightly lower than the seawater ratio of 1.8 (Seinfeld and Pandis, 2006). Cl$^-$ depletion is commonly observed for ambient SSA, and is largely attributed to atmospheric aging processes. Cl$^-$ depleted nascent SSA, as observed here, has also previously been reported to indicate that chloride is fractionated in seawater depending on the seawater composition or that Cl is evaporated during SSA production (Schwier et al., 2017). Wave chamber experiments identified an externally mixed C and O containing particle type, which contained inorganic elements such as S, Na, Mg, Ca and K, but not

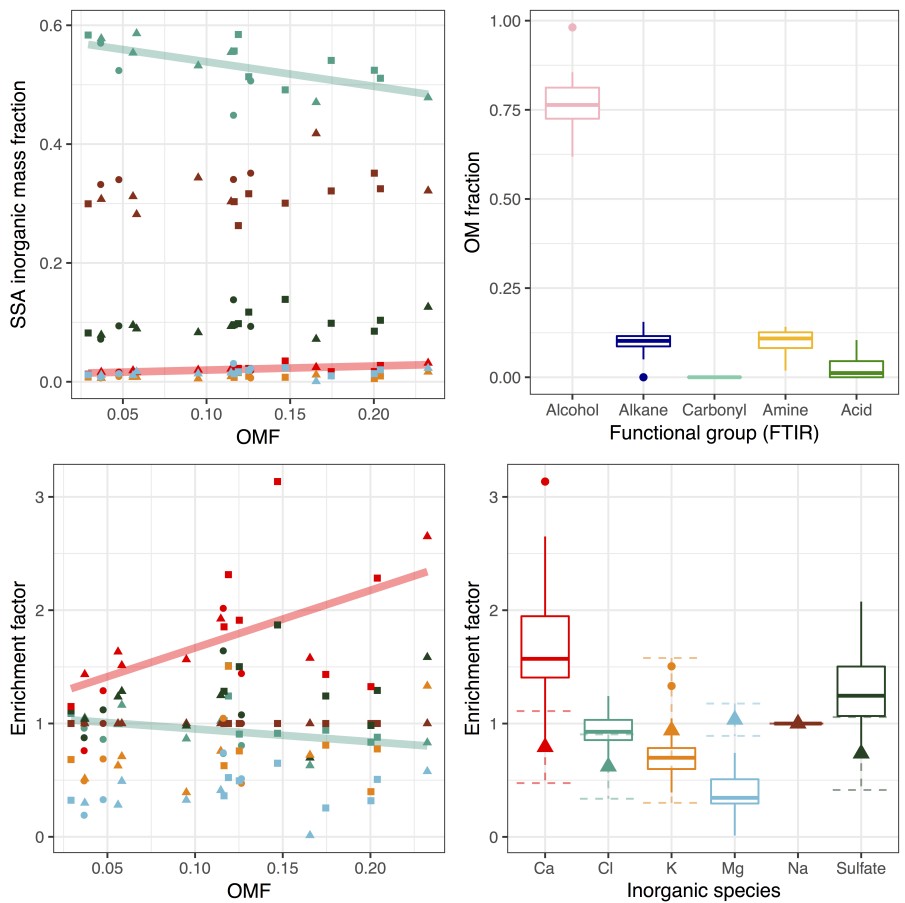

**Figure 7.** Inorganic mass fraction (top left) and enrichment factor (bottom) versus organic mass fraction (bottom left) measured from IBA and FTIR analysis of filter samples. Enrichment factors of inorganic species are with respect to laboratory prepared sea water and presented with respect to $Na^+$. Left hand plots show linear trends for species with statistically significant correlations (Cl fraction displayed $R^2$ of 0.34, p-value $< 0.01$, and slope of $-0.9 \pm 0.3$, and Ca fraction displayed $R^2$ of 0.42, p-value $< 0.01$, and slope of $7 \pm 2$). Shapes indicate water sample depth. Colours in the left panels correspond to inorganic species indicated in the bottom right panel. Stars in bottom right plot represent the mean EF from TEM-EDS measurements of SSA generated from laboratory seawater, dotted error bars show standard deviation in the mean. Contribution to organic mass from functional groups measured by FTIR shown top right. Boxes extend from the 25th to the 75th percentile, with the line showing the median, crosses show measurements outside of the 95% confidence interval in the median.

Cl (Ault et al., 2013a) and the presence of this particle type could decrease the overall Cl contribution. Enrichment of $Ca^{2+}$ is consistent with other nascent SSA chamber experiments, a proposed explanation for this is the complexing of $Ca^{2+}$ with carbonate ions. The presence of carbonate would potentially be detected in the $OVF_{NV}$ from TDMA measurements and could therefore provide an explanation for the over prediction observed in Fig. 6. Alternatively $Ca^{2+}$ could be in a complex with

5    organics, Salter et al. (2016) concluded that if this were the case it would be with a minor amount of organic material.





Alcohol functional groups contributed $77 \pm 8$ % of the SSA OM, alkanes $10 \pm 4$ %, amines $10 \pm 3$ % and carboxylic acid groups $3 \pm 3$ % ($mean \pm sd$). The make up of organics across the samples was relatively constant, as depicted by the ranges shown in Fig. 7. The ratio of alkane to hydroxyl (alcohol) functional groups indicates whether the organic fraction is aliphatic/ lipid like (high ratio) or more oxidised/carbohydrate like (low ratio). The nascent SSA generated during SOAP had alkane

to hydroxyl ratios ranging from 0.06 tot 0.25, with a mean of 0.14, which are very low values for non-oligotrophic waters, suggesting that the SSA was enriched in carbohydrates. For some context Frossard et al. (2014) reported average ratios of $0.34 \pm 0.21$ for non-productive waters and $0.93 \pm 0.41$ for productive waters. Chlorophyll-a concentrations for productive waters reported in Frossard et al. (2014), extend to 10 $\mu g\ L$, far above those observed in this study. It is also worth noting that the alkane to hydroxyl ratios were lowest during blooms 1 and 3, $0.12 \pm 0.04$ and $0.11 \pm 0.04$, respectively and highest

outside of phytoplankton blooms $0.2 \pm 0.05$. These results suggest that the SSA from phytoplankton blooms is enriched in carbohydrate-like organics, more so than the less biologically active regions.

Ethanol growth factors measured using the UFO-TDMA for preselected 50 nm diameter SSA were $1.22 \pm 0.02$ (mean $\pm$ sd) and were largely invariable for all of the water samples examined. Applying the ZSR assumption with an organic growth factor of 1.5 and a sea salt growth factor of 1, the measured ethanol growth factors correspond to moderately oxidised organic volume

fractions averaging $35 \pm 5\%$. The variability due to SSA diameter in the ethanol growth factors measured at 15 to 50 nm were all within experimental error once a correction for the Kelvin effect was applied. There were no significant correlations with the 50 nm ethanol growth factor and any of the water or particle phase variables measured. The ethanol growth factor of volatilised SSA (for sample U7520) was $1.03 \pm 0.03$ at 200 °C, and averaged $1.01 \pm 0.03$ between 250 and 400 °C, suggesting that the component contributing to ethanol growth was largely semi-volatile. The component that contributed to ethanol growth was

more constant than the OVF$_{SV}$ measured using the VH-TDMA, suggesting that it could have been a subset of the total volatile organic component.

SSA was generated from a number of CTD water samples at two different depths, one in the mixed layer and one deep water sample as shown in Table S1. Notable differences were that OVF$_{SV}$ and OVF$_{NV}$ were over 1.4 times greater in the mixed layer than in deep water and the OMF was up to 1.8 times higher. $Ca^{2+}$ and $SO_4^{2-}$ were 1.7 and 1.3 times higher in the mixed layer,

respectively. $Na^+$ was slightly lower (a factor of 0.9) and alkanes in the aerosol phase were up to 2.4 times higher. The effect of depth was only looked at for three CTD samples, therefore the representativeness of the above values should be treated with caution, however the relative increase in SSA organic fractions, calcium and alkanes is consistent with higher biological activity towards the ocean surface.

### 3.4   SSA water uptake

The HGFs observed for SSA generated from both laboratory sea salt and natural seawater samples showed up to 3 externally mixed HGF modes (Fig. S5). The first natural seawater SSA HGF mode averaged $1.89 \pm 0.07$ and contributed a number fraction of $0.8 \pm 0.12$ for 50 nm diameter SSA. The second mode displayed an average HGF of $2.04 \pm 0.09$ and contributed a number fraction of $0.2 \pm 0.1$. The third HGF mode was sporadically observed during SOAP measurements at 50 and 100 nm diameters (observed during 4 samples), but when present contributed a number fraction of 0.01 to 0.06 and displayed an average HGF





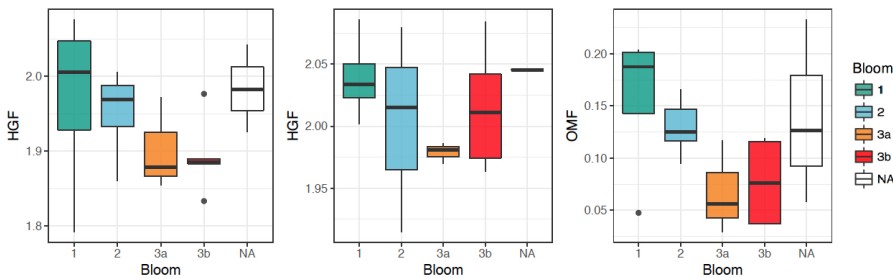

**Figure 8.** Ambient HGF (left), heated HGF (middle) and OMF (right) by bloom. HGFs measured using VH-TDMA, OMF measured from FTIR and IBA analysis of filter samples. Note that NA refers to measurements taken outside of an identified bloom.

of $2.25 \pm 0.02$. The fraction of the second HGF mode at 50 nm correlated with the proportion of lognormal mode 3 ($R^2$ of 0.39, p-value $< 0.01$, and slope of $0.87 \pm 0.3$), suggesting that log-normal mode 3 is made up of particles from the second HGF mode. Similarly, the first two log-normal modes have similar hygroscopicities and are related to the first HGF mode. The presence of externally mixed HGFs for the natural and laboratory seawater samples suggests that the composition and/or
morphology is different between log-normal modes 1-2 and mode 3.

The shape corrected 50 nm ambient nascent SSA HGF averaged $1.94 \pm 0.08$ (mean $\pm$ sd) across all samples, with individual samples ranging from $1.79 \pm 0.05$ to $2.08 \pm 0.06$ as shown in Fig. 5. Heated 50 nm HGFs averaged $2.02 \pm 0.05$ across all samples, with individual samples ranging from $1.91 \pm 0.06$ to $2.09 \pm 0.06$. Particularly interesting is the distribution of HGFs throughout the voyage as shown in Fig. 8, both nascent SSA HGFs and OMFs were highest for bloom 1 on average, and
decreased for subsequent blooms. The SSA HGF after heating is approximately 0.1 higher than that from the ambient HGF, which is a similar change in HGF as observed for laboratory sea salt samples, and is likely to be largely due to the evaporation of hydrates. The relationship between HGF and OVF or OMF observed here is not consistent with that expected from the ZSR assumption i.e. full solubility of organic components. Conventional ZSR mixing would suggest the organic fraction and water uptake would be inversely proportional to each other, because of the presence of a less hygroscopic organic component. Even
when a HGF of 1.6 is assumed for the organic component, the trend in HGF is not consistent between measured and ZSR modelled HGFs (Fig. 9). Deviations between the ZSR model and the measured data begin to become pronounced at OVFs greater than 0.4. It should also be noted that the measured HGFs show a large sample to sample variability.

A threshold organic fraction, beyond which the droplet diameter is enhanced, has previously been observed for fatty acids, and is related to changes in the droplet surface tension (Forestieri et al., 2018; Ruehl and Wilson, 2014; Ruehl et al., 2016).
A buffered response of SSA hygroscopicity to OVF has also been previously reported (Ovadnevaite et al., 2011a; Collins et al., 2016; Forestieri et al., 2018) and is thought to be linked to surface active organics (e.g. fatty acids). Given the observed combination of a low alkane to hydroxyl ratio and an apparent non-soluble organic component, lipopolysaccharides (LPS) could present a reasonable candidate for the composition of the organic component. LPS have previously been identified as an important component in primary marine aerosol (Cochran et al., 2017; Estillore et al., 2017; Facchini et al., 2008; Bikkina
et al., 2019).





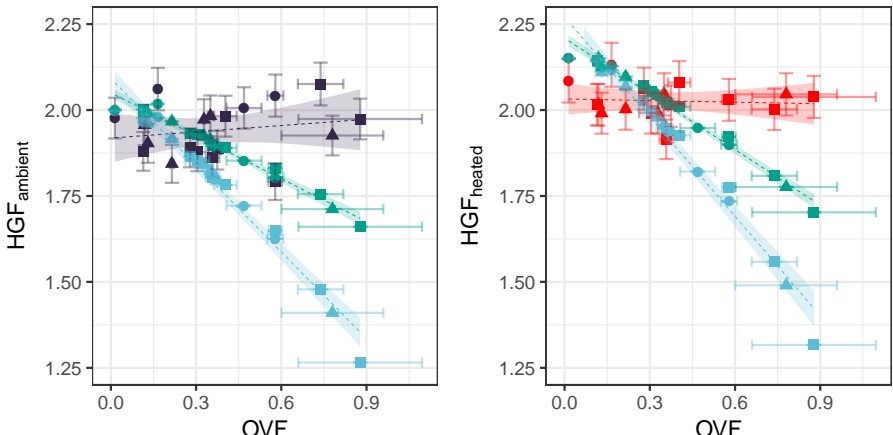

**Figure 9.** The measured HGF of 50 nm diameter SSA at ambient temperature (left) and heated to 255 - 400 °C (right) as a function of OVF. HGF modelled using ZSR assumption shown assuming an organic HGF of 1 (blue) and an organic HGF of 1.6 (green) shown alongside measured HGFs (ambient in dark blue, heated in red). Linear fit to the measured and modelled data indicated by dashed line, shading represents the 95% confident interval in the linear fit.

The deliquescence relative humidity was measured for the Workboat 9 seawater sample at 69% RH (Fig. 10), notably lower than that observed for NaCl/sea salt, ∼73.5 % (Zieger et al., 2017). SSA generated from Workboat 9 seawater displayed an ambient HGF of $1.84 \pm 0.06$, a heated HGF of $1.94 \pm 0.06$, an organic mass fraction of approximately 3% and an organic volume fraction of approximately 21%. The alkane to hydroxyl ratio was 0.14. The DRH observed here is consistent with observed organic sea salt mixtures in the literature, for example a 2:1 mass ratio mixture of NaCl to glucose resulted in a 100nm DRH of $69.2 \pm 1.5$ % and the mixture had a hygroscopcity ($\kappa$) of 0.8 (Estillore et al., 2017).

## 4    Discussion

### 4.1    SSA organic enrichment

The organic enrichment of SSA was examined using the Chl $a$ based emissions scheme suggested by Gantt et al. (2011), the OCEANFILMS-1 emissions scheme (Burrows et al., 2014) and the OCEANFILMS-2 emissions scheme which allows for the co-adsorption of polysaccharides (Burrows et al., 2016). The relationship between SSA OMF estimated using Chl $a$ (Gantt et al., 2011) and the measured SSA OMF is quite scattered ($R^2$ of 0.17), as shown in Fig. 11. The nature of the sampling method used in this study, 23 spot samples taken over an 18 day period, was not favourable for the use of Chl $a$ as a marker for SSA organics. The enrichment of SSA is not just a product of phytoplankton biomass, which is largely what is measured by Chl $a$, but more likely due to the demise of phytoplankton communities and the resulting release of a range of organic material, some of which is available for transfer into the aerosol phase. Chl $a$ is best used as marker over much longer time scales, of the order of weeks to months, for example to describe the seasonality in SSA organic enrichment. The OCEANFILMS-1





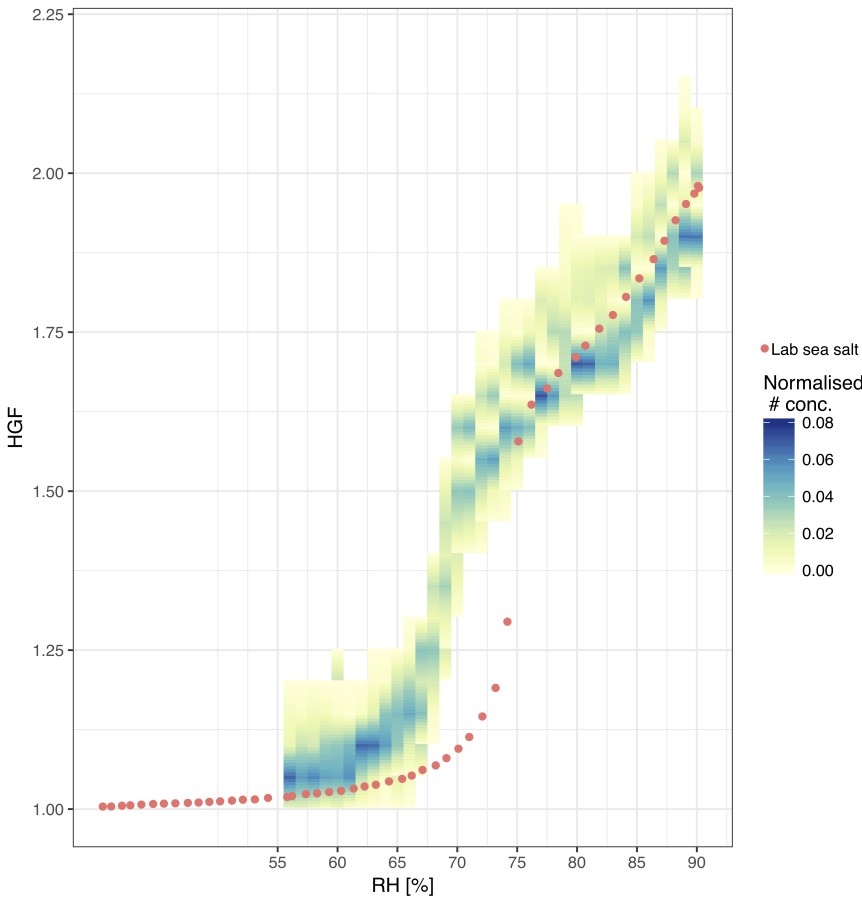

**Figure 10.** Deliquescence of SSA generated from SOAP sea water sample (Workboat 9) shown in yellow to blue colour scale. Laboratory sea salt deliquescence curve shown with circles.

model improves on the scatter of modelled organic fraction compared to the Chl $a$ model ($R^2$ of 0.3), however the magnitude of the modelled OMF is low, which is likely due to the under-representation of more soluble DOC, such as polysaccharides. OCEANFILMS-2 includes the co-adsorption of polysaccharides and reproduces the SSA OMF reasonably well ($R^2$ of 0.44). It is worth noting that both OCEANFILMS models over predict when the organic fraction is low, OMF $< 0.05$.

5    The organic macromolecular classes associated with the SSA were determined from the functional group composition using the conversions outlined in Burrows et al. (2014), which relates the concentration of each molecular class to a weighted sum of the functional group concentrations. The high proportion of alcohol functional groups from observations resulted in a similarly high contribution from the polysaccharide-like molecular class, with an average contribution of $0.72 \pm 0.06$ to the total OMF. The conversion between functional groups and macromolecular classes is based on the properties of characteristic molecules, for example cholesterol and simple sugars, and is therefore not a perfect representation of the marine environment. There was no apparent change in the distribution of molecular classes/functional groups with the organic mass fraction, as shown




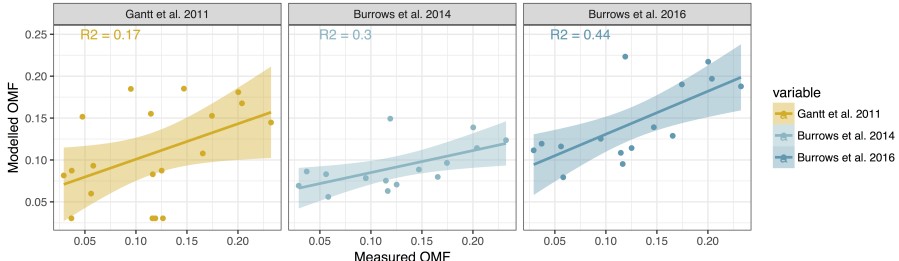

**Figure 11.** OMF modelled using parameterisation from Gantt et al. (2011) (left), OCEANFILMS-1 (middle) and OCEANFILMS-2 (right) compared to that measured in this study. Lines show the linear fit between modelled and measured OMF, shading shows the 95% confidence interval in the linear model. Fit parameters for Gantt et al. 2011: $R^2$ of 0.17, p-value $< 0.05$, and slope of $0.4 \pm 0.2$. Fit parameters for OCEANFILMS-1: $R^2$ of 0.3, p-value $< 0.05$, and slope of $0.3 \pm 0.1$. Fit parameters for OCEANFILMS-2: $R^2$ of 0.44, p-value $< 0.01$, and slope of $0.5 \pm 0.1$.

in Fig. 12, despite an increase in seawater polysaccharides and proteins for samples with higher OMF (see Fig. S1). These results might suggest that the SSA organics were bound in a similar molecule or complex, which was uniform regardless of the organic mass fraction. As expected OCEANFILMS-1 underestimated the proportion of polysaccharides and over estimated the proportion of lipids, as shown in Fig. 12. OCEANFILMS-2 displayed an improved representation of polysaccharides, however

the proportion was still underestimated with an average contribution of $0.39 \pm 0.06$ to the total OMF. It is worth noting that the processed molecular class was not computed from FTIR measurements because the functional group composition is so similar to the polysaccharide-like class. The contribution of the processed class to the SSA organic fraction from both OCEANFILMS models was very low (Fig. 12).

     Over prediction of alkane to hydroxyl ratios for particularly clean marine measurements is a known issue for OCEANFILMS-

2, and broadening the model to different saccharides with varying molecular weights has been identified for future research (Burrows et al., 2016). Particularly of interest here is research into the interaction of surfactants with divalent cations, which have been observed to impact the orientation of surfactant head groups, and thus the surface pressure, for modelled surfactant salt systems (Adams et al., 2016; Casillas-Ituarte et al., 2010; Casper et al., 2016). In particular $Ca^{2+}$ has been observed to form particularly stable complexes, bridging neighbouring surfactant molecules, and having a condensation effect on the monolayer

(Casper et al., 2016). The enhanced enrichment of $Ca^{2+}$ with higher OMF observed herein may therefore be associated with the complexation with surfactants, in which case the enrichment of $Ca^{2+}$ could influence the organic enrichment and the SSA water uptake.

     OCEANFILMS does improve the prediction of organic enrichment from seawater parameters, relative to Chl $a$ based models. Limitations remain in the implementation of OCEANFILMS, which requires the availability of surface water concentrations

for the five macromolecular classes, which are generally generated using biogeochemical modelling. In addition there are remaining uncertainties as to the global applicability of the model organics applied in OCEANFILMS, in regions with different phytoplankton populations for example. Further broadening and/or refining the organics of interest in OCEANFILMS is likely





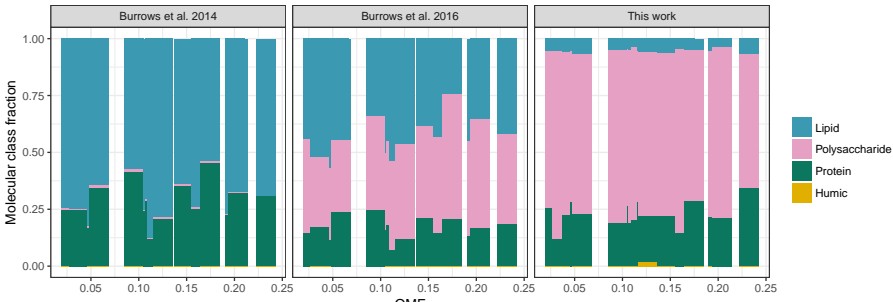

**Figure 12.** Measured organic composition inferred from functional groups (right) and modelled organic composition from OCEANFILMS (OCEANFILMS-1 left panel, OCEANFILMS-2 middle panel).

to hinder its application. Similar issues of global applicability are also present for Chl $a$ based estimates of OMF. Chl $a$ is, however, globally observed at a daily timescale via satellite and is extremely important for large scale simulations of SSA organic enrichment. Measurements over the sparsely observed Southern and South Pacific Oceans, such as those reported herein, are important to constrain emission schemes developed in other parts of the world, particularly given the importance of

SSA in this region.

## 4.2 SSA water uptake

The compressed film model was run assuming organics with a range of molecular areas and molar volumes to test the sensitivity of HGF to these variables. The critical molecular surface areas that minimised the error in the modelled HGF were most commonly 35 - 45 square angstroms for both ambient and heated samples, which is consistent with laboratory observations

of marine salts mixed with a phospholipid found in the SML (DPPC) (Casper et al., 2016). Molar volumes of less than 10 $\mathrm{cm}^3 \, \mathrm{mol}^{-1}$ were typically observed. This is well below the threshold observed by Forestieri et al. (2018) to induce surface tension effects significantly different than those when a surface tension of water is assumed. It should be noted that significant sample to sample variability was observed in the fitted molecular volume and surface area.

The proportion of SSA organics at the particle surface was tested assuming that partitioning occurs on the basis of the organic

molecular classes as computed from the distribution of functional groups or from the OCEANFILMS model. Four cases were tested assuming the following molecular classes partitioned to the surface:

- lipids,

- lipids and polysaccharides,

- lipids polysaccharides and processed organics, and

- all organics.



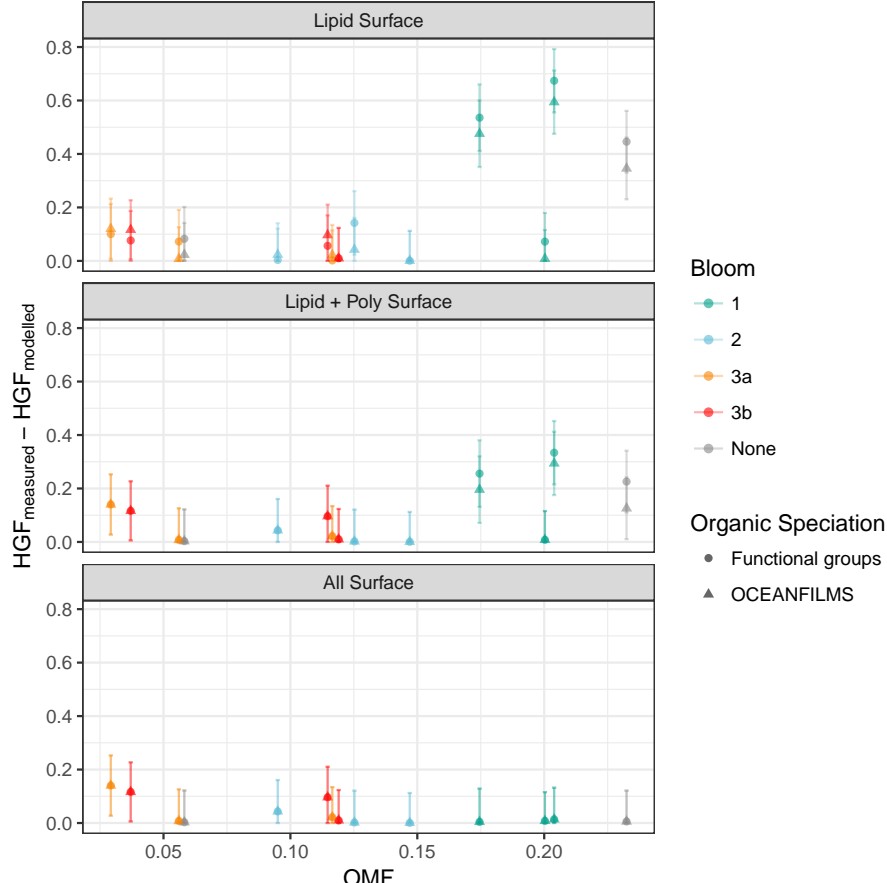

**Figure 13.** Compressed film model ambient HGF error (absolute value of measured minus modelled HGF) for lipids partitioned to the surface (top), lipids and polysaccharides partitioned to the surface (2nd panel from top), lipids polysaccharides and processed organics partitioned to the surface (2nd panel from bottom) and all organics partitioned to the surface (bottom). Organic speciation derived from FTIR measurements (circles) and from OCEANFILMS model (triangles). Compressed film model $A_0$ and $V_{org}$ values were determined by those that resulted in the best fit between observed and modelled HGF for each sample.

The resulting error in the HGF given by compressed film model is shown in Fig. 13 and Fig. S4. The most notable feature is that for both ambient (Fig. 13) and heated measurements (Fig. S4) the error in the modelled HGF is significantly reduced at high OVFs when all of the organics are partitioned to the surface. At low OVFs the sensitivity to partitioning of organics is low. Note that Fig. 13 and Fig. S4 display the error in the HGF as a function of OMF, very similar results are achieved 5   when the error is reported as a function of Aitken mode OVF. The results presented here suggest that at high OVFs all of the organics, including the proteinaceous components, partition to the surface of the SSA. The water uptake results coupled with the relatively invariable SSA organic molecular classes/functional groups could suggest that the organic composition





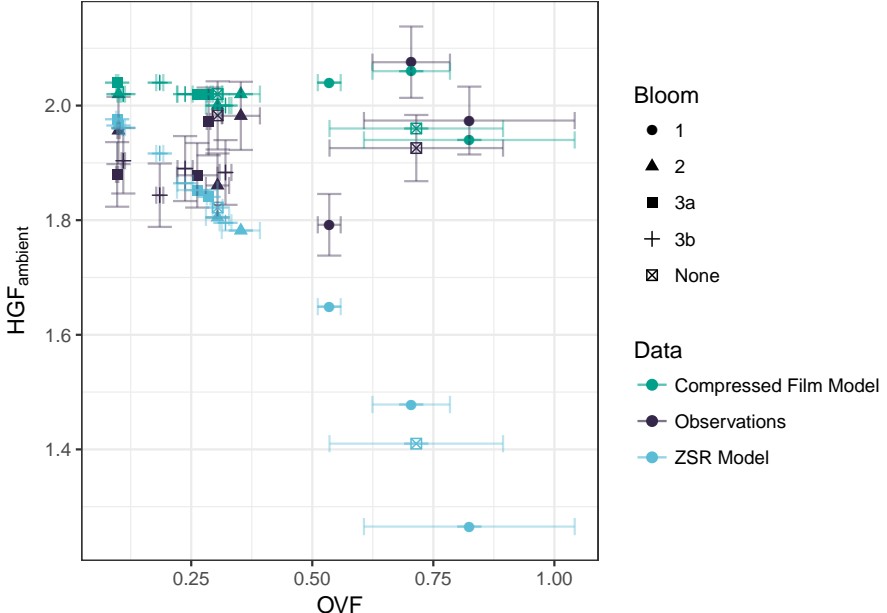

**Figure 14.** Ambient HGF modelled using compressed film model (green) as a function of OVF, modelled using the ZSR assumption (light blue) and observed (dark blue). Compressed film model output is for case where all organics are partitioned to the particle surface. Compressed film model A0 and Vorg values were determined by those that reduced the overall error between observed and modelled HGF for all of the samples.

is relatively similar for all samples e.g. LPS, but surface tension effects are observed when the OVF is high enough for a monolayer to form.

Directly comparing the HGFs modelled using the compressed film model and those modelled using the ZSR assumption, as in Fig. 14, highlights the contribution of surface tension to the observed SSA HGF. The surface tension effects on HGF observed

at high organic volume fractions are approximately equivalent to the reduction in HGF predicted by the ZSR assumption, i.e. by Raoult's Law. The role of surface tension is important for prediction of CCN, in addition OVFs inferred in studies using water uptake techniques, using the ZSR assumption, have often been lower than those measured using more direct analyses of SSA chemical composition. For example, water uptake measurements made by Fuentes et al. (2011) and Modini et al. (2010a) yielded Aitken mode SSA organic volume fraction of up to 40%, while measurements with similar seawater Chl $a$ and DOC

concentrations measured OVFs of up to 80% (Keene et al., 2007; Facchini et al., 2008; Prather et al., 2013). A contribution to this apparent discrepancy could be the under-prediction of OVF due to the presence of surface active organics, and the resulting depression of surface tension. The contribution of surface tension to the discrepancy between reported observations is unknown as there is also likely to be a contribution from variability in the marine organics and resulting SSA enrichment.

The impact of the reduction in SSA surface tension should be measured by what influence it has on the overall CCN

concentrations. The CCN concentrations predicted from the compressed film model, with all organics partitioned to the surface,





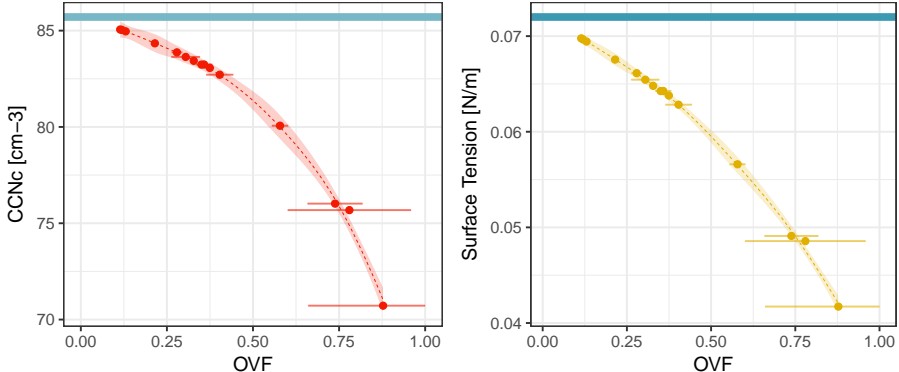

**Figure 15.** Modelled CCN concentrations (left) and surface tension (right) as a function of OVF. The CCN concentrations are modelled assuming an SSA size distribution with number concentration of $100\,\mathrm{cm^{-3}}$, a mean diameter of 160 nm and a geometric standard deviation of 2.6. Blue line shows the CCN concentration predicted using the compressed film model with all organics partitioned to the surface and with an assumed Dcrit of 50 nm, red points show the predicted CCN concentrations using the ZSR assumption. The surface tension required to obtain a Dcrit of 50 nm is represented with yellow circles (right), and the blue line shows the surface tension of water. The dashed lines are a LOESS fit to the data, and the shaded regions are a 95% confidence interval in the fit.

were compared with those computed using the ZSR assumption, when both models were applied to a theoretical SSA size distribution (Fig. 15). The compressed film model was applied assuming that the critical diameter was equal to the VH-TDMA pre-selected particle diameter (50 nm), and the $\kappa$-Köhler equation was used to calculate CCN concentrations from the ZSR modelled HGFs using the same critical SS as in the compressed film model. Setting up the calculation as described above gives

a comparison of the ZSR CCN concentration, which varied with OVF, to the compressed film CCN, which was stable across all samples. In addition the surface tension required to obtain the compressed film CCN was also calculated. The decrease in the surface tension was up to $30\,\mathrm{mN\,m^{-1}}$, which is a large reduction in surface tension, similar to that modelled for binary SSA proxies by Forestieri et al. (2018). The increase in CCN associated with the surface tension modification was up to 10 $\mathrm{cm^{-3}}$, which represents a 17% increase in CCN concentrations from the ZSR approach. Translated into cloud droplet nuclei

concentrations, a 17% increase would have a potentially large impact on the cloud fraction, cloud liquid water content and cloud radiative forcing (Rosenfeld et al., 2019). The results presented here suggest that the inclusion of surface tension effects for SSA could improve the representation of SSA CCN in atmospheric modelling.

## 5    Conclusions

Chamber measurements of primary marine aerosol generated from 23 seawater samples collected across three phytoplankton
blooms tracked during the 23 day SOAP voyage over the Chatham Rise (east of New Zealand) are examined in this study. The SSA was an internal mixture of sea salt and organics. Volatility measurements at a preselected particle mobility diameter of 50 nm indicated that the SSA had an organic volume fraction of up to 0.87, with an average of $0.36 \pm 0.24$, largely made





up of a refractory component. Filter measurements of PM for diameters less than approximately 1 $\mu m$ were analysed for the concentration of organic functional groups using FTIR, and the concentration of inorganic species was determined using IBA. The organic mass fractions ranged from 0.03 to 0.23, and had a large proportion of hydroxyl functional groups which, along with very low alkane to hydroxyl ratios, suggests a polysaccharide rich, less aliphatic organic species. $Ca^{2+}$ was observed to

be 1.7 times higher in the aerosol phase than in seawater, which is consistent with other primary marine aerosol studies. A possible explanation for this is that $Ca^{2+}$ complexes with organics in the SML, which is supported by the correlation of the $Ca^{2+}$ enrichment factor with SSA OMF.

The SSA organic fraction displayed a scattered correlation with chlorophyll-a, consistent with previous studies which show that chlorophyll-a is best used to correlate SSA organic enrichment over larger spatial scales and at temporal scales of the

order of months. The OCEANFILMS model provided an improved representation of the SSA organic fraction observed in this study, in particular when the co-adsorption of polysaccharides was included i.e. OCEANFILMS-2 was applied. High hydroxyl to OMF fractions were observed from FTIR measurements, which translated to large estimated contributions from the polysaccharide like molecular class of $0.72 \pm 0.06$. OCEANFILMS-2 underestimated the contribution from the polysaccharide-like molecular class ($0.39 \pm 0.06$). Further work on representing the adsorption of polysaccharide species is required.

Water uptake measurements revealed that the SSA hygroscopicity was largely invariable with the organic mass fraction, with HGFs averaging $1.93 \pm 0.08$. The observed HGFs deviated from the regularly used water uptake mixing rule, the ZSR assumption, particularly during B1 when organic volume fractions were above 0.4. The representation of hygroscopicity was drastically improved when the compressed film model was applied. The error in modelled water uptake was minimised when all of the organics were partitioned to the surface in the compressed film model. Surface tension effects could be one reason

for the discrepancies that have been observed between SSA organic fractions estimated using water uptake and those estimated using other methods. A decrease in CCN concentrations of up to 17 % was estimated when the ZSR assumption was used, and therefore the inclusion of SSA surface tensions effects could improve the representation of SSA CCN in atmospheric modelling.

The SSA organics showed consistently low alkane to hydroxyl ratios, even in relatively productive waters with high SSA

organic fractions, and surface tension effects. These results could indicate that the source and structure of the SSA organics was largely consistent throughout the voyage, for example made up of lipopolysaccharides, which have previously been identified as an important component in primary marine aerosols. These measurements provide valuable comparison with observations for models of SSA organic enrichment and water uptake. Constraints on emissions and process models for this region are of particular importance because existing measurements are sparse and it is a region for which SSA has been observed to make

up a large contribution to CCN.

*Data availability.* The data are available through the World Data Center PANGAEA (https://www.pangaea.de/) in the near future. The nascent SSA composition, water uptake and volatility data is available in the supplementary material. Further data and information are available by request to the corresponding author or the voyage leader Cliff Law (cliff.law@niwa.co.nz).



*Author contributions.* The SOAP campaign was led and coordinated by CSL and MJH. CSL led the ocean biogeochemistry work programme and MJH led the atmospheric work programme. LTC, MDM, PV, MJH and CSL made measurements during the SOAP voyage. GO developed fatty acids and Alkane techniques and analysed samples. KS collected workboat samples, conducted Chl-a sampling and optical microscopy on phytoplankton species. TB sampled and analysed High Molecular Weight Sugars and Proteins. RLM and LMR performed FTIR analysis
on SSA filter samples. ES and LTC performed IBA analysis on filter samples. LTC led analysis and interpretation of data, with input from all authors (ZR in particular) and the manuscript production, with input from all authors.

*Competing interests.* The authors declare that they have no conflict of interest.

*Acknowledgements.* We acknowledge the invaluable assistance of the captain, officers and crew of the R/V Tangaroa. We thank Gus Olivares and Nick Talbot collection and analysis of samples. In addition we would like to thank researchers, in particular Robin Modini, for providing
time and expertise in running FTIR analysis on these samples. FTIR analyses were also supported by National Science Foundation grant NSF AGS-1013423. The SOAP study was supported by funding from NIWA's Climate and Atmosphere Research Programme 3 – Role of the oceans (2015/16 SCI), and VOC and CCN measurements were supported by CSIRO's Capability Development Fund. Petri Vaattovaara's participation was supported by EU COST action 735, the Academy of Finland, through the Centre of Excellence and via a Finnish Academy visiting grant, no. 136841. We would like to thank AINSE Limited for providing financial assistance (Award - ALNGRA13048) to enable
the Ion Beam Analysis of elemental composition of marine aerosol filters collected during the SOAP voyage. Data analysis was supported by ARC Discovery grant DP150101649. Some of the data reported in this paper were obtained at the Central Analytical Research Facility (CARF) operated by the Institute for Future Environments (QUT). Access to CARF is supported by generous funding from the Science and Engineering Faculty (QUT).



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
