# Peer review of "Sea spray aerosol organic enrichment, water uptake and surface tension effects"

_Atmospheric Chemistry and Physics, 2019_

## Referee Comment (RC1) · Anonymous Referee #1 · 8 Nov 2019

The authors present observations of particle evaporation and sub-saturated hygroscopic growth for particles produced from bubbling air through seawater that was collected from a number of locations around New Zealand. They compare these observations with concurrent measurements of the particle composition and with the seawater composition. They use their evaporation measurements to infer organic content of the sampled particles. Among other findings, they observe the particles to have hygroscopic growth factors greater than expected given the large inferred organic content for some of the samples. They attribute this to surface tension depression. I have a number of questions and suggestions for clarification, and have some concerns about their interpretation of the sub-saturated hygroscopic growth. This paper should ultimately be publishable, but I think there are a number of aspects that require clarification first.

[Figure]

P1/L9: I suggest "the Aitkin mode" be specified as a size range.

P1/L16: The particle size should be stated.

P2/L1: The influence of surface partitioning is quite small on sub-saturated hygroscopic growth. The impact of surface partitioning, and surface tension depression, becomes much more important at very high RH, near 100%. If the authors want to highlight this as a reason for their observations, they need to perform calculations within the main text that illustrate the importance of this effect. Simply stating that surface partitioning can explain the results is insufficient. The discussion on Page 20, which is in reference to various CCN measurements and not sub-saturated hygroscopic growth measurements, is not directly relevant. There needs to be a clear discussion of the impact at the conditions of the measurements. I suggest this sentence be deleted unless it can be backed up with appropriate calculations. The details regarding the compressed film model and how it was used (Page 24) are not sufficiently clear to allow a reader to understand what specifically was done.

P3/L7: I suggest it would be better to refer to the OVF values determined from hygroscopicity measurements as "derived" rather than "observed."

P3/L11: I suggest "volatilizable" is more appropriate than "volatile" to describe the organic material in SSA. As the material exists in the condensed phase, it is not exactly "volatile."

P3/L13: As the location is given for the Modini paper, it seems appropriate to give it for the Quinn paper as well.

P3/L15: I do not see how the Modini paper here concluded that there was not a non-volatile organic fraction. Modini et al. characterized only the volatility and hygroscopicity. They did not characterize organic components. Thus, they would not be able to directly address the question of non-volatile, residual organic compounds.

P3/L20: As fatty acids tend to have long hydrocarbon tails, their presence is not necessarily consistent with a large hydroxyl fraction, as stated. This is noted in the next sentence. I suggest these are better aligned.

P4/L13: I do not follow how the studies cited in this sentence are "chamber" studies. I don't think any of these are actually chamber studies. Also, suppressed relative to what? Relative to ZSR? This would then conflict with the first part of the sentence. This could be clearer.

P4/L16: It would be useful if the authors would more clearly distinguish between sub-saturated and super-saturated measurements here, as these can be quite different in their response to added organics.

P4/L14: Technically, the Ovadnevaite paper is not on SSA. It is on secondary particles in the marine environment. These should be distinguished, as the composition is very different.

Introduction in General: As the composition of the organic material may (and likely does) vary with particle size, it might be helpful if the authors were to be as explicit as possible in stating the size range of the measurements when they refer to different studies. For example, are the measurements the total submicron? For just smaller particles? The authors might also consider adding further discussion regarding what the literature suggests about variability in the OA chemical composition with size.

P6: Particle generation: What concerns do the authors have regarding the representativeness of their size distributions obtained from their particle generation method, and how this might influence particle composition? The observed particle distributions (Fig. 4) differ notably from measurements of sea spray particles from breaking waves (e.g. Prather et al., PNAS, 2013) or estimated from multi-mode fitting of ambient distributions (e.g. Quinn et al., Nat Geosci., 2017; Saliba et al., PNAS, 2019). Differences in size distribution can be indicative of differences in composition. This issue needs to be explicitly discussed, including discussion of potential biases that might result.

Fig. S2: These are not exactly "volatility profiles" as stated in the main text, but instead a comparison of the volatile fraction for sea spray versus sea salt. A volatility profile would be a graph of volatile fraction or fraction remaining versus temperature.

P7/L15: The Vaattovaara et al. (2005) paper indicates that there is negligible growth for 10 nm inorganic particles, but that growth of larger particles in ethanol vapour is not negligible. Furrhter, that paper did not consider sodium chloride. Has it been demonstrated that this method works for particles more representative of those sampled here, that is do sodium chloride (or sea salt) particles not grow? This would be helpful to place uncertainty bounds on the authors measurements. If this has not been demonstrated, how do the authors derive an uncertainty?

Methods: There is a general lack of discussion of uncertainties. Such discussion would be welcome (aka is really needed).

P8/L17: The TDMAinv method does not account for multiply charged particles. Are these a concern? Based on the size distributions shown, I would think they would be.

P8/L22: I do not find it clear how the volatility of hydrates is accounted for, nor how consideration of the hydrate proportion yields the organic fraction. If much of the organic fraction is truly non-volatile, wouldn't this method fail? Or if organic material chars to become non-volatile? Also, wouldn't this method fail if organics also evaporate between 200C and 400C? Organic volatility tends to be a continuum. Thus, one might not expect a bimodal distribution, as assumed here. How do the authors justify this assumption regarding organic volatility? It may be reasonable, but requires justification. (Note: if something is non-volatile, then it doesn't evaporate. Thus, the "non-volatile" organics indicated here are not non-volatile, but very low volatility. I strongly suggest the authors adopt a more precise language.)

Eqn. 2 relationship to OVF_tot: I think that these relationships could be stated more clearly. It is not clear, at least to me, why the total OVF would be 1-f (P9/L3). If the OVF_SV = 0, then in Eqn. 2 the value of f is by definition unity. The OVF_tot

would then be 1-1 = 0. But, this wouldn't account for evaporation above 400C. So it is unclear to me how this works. It is similarly unclear how the authors end up with OVF_tot values up to 0.9 (Fig. 6) when the max VF values in Fig. S2 reach only 0.23. Perhaps I am misunderstanding, but I find the discussion here to be unclear, making it difficult to really understand how the OVF_tot values were determined. I suggest revision is needed. I strongly encourage the authors to include graphs of VF (or VFR) versus temperature so that the reader can clearly see what must be a step change after ∼400C indicative of evaporation of "non-volatile" components. There must be huge differences between the samples with OVF_tot = 0.9 and those with small values. It would be useful to the reader to see these.

P15/L5: The authors need to provide much more detail here regarding how they apportion things to the different modes. There must ultimately be full consistency with all the measurements. How do they decide how much of the volatile and "non-volatile" material goes between the three modes that overlap? Does this add up appropriately to, hypothetically, reproduce the observations? It is also not clear whether the authors assumptions allow for any salts in modes 1-3, or whether these are limited to mode 4. As written, they only indicate salts in mode 4. If this is the case, then the assumptions here are inconsistent with the VF observations. There must be a salt component at 50 nm, based on their interpretation, and thus there must be some salt in either Mode 1, 2, or 3. Additionally, the authors point the reader to Section 3.4 to justify their split, but it is not evident after reading Section 3.4 exactly how they made this determination. They need to be more explicit here and (i) fully justify their choices while (ii) demonstrating the internal consistency. As best I can tell from the range of sizes considered, the authors do not have an independent constraint on the composition of Mode 4 since it contributes negligibly to the number concentration below 150 nm, although will have a large influence on the overall mass.

OVF correlations: What does the slope of OVF versus some seawater metric (e.g. concentration of alkanes) mean? These are reported, but the meaning is not clear as

the OVF is a fraction of the total PM. Also, shouldn't these slopes have units?

P15/L10: An assumption of an organic density of 1.1 g/cm3 seems at odds with the determination that species such as saccharides dominated the composition (based on the large hydroxyl fraction). Can this be further justified? Citation of Modini is insufficient, as that paper simply assumed 1.1 g/cm3 based on Keene et al. (2007) and thus is not an independent determination. Then, Keene et al. (2007) do not actually determine this, but state it is estimated based on Schkolnik et al. (2006). The title of Schkolnik et al. (2006) is "Constraining the density and complex refractive index of elemental and organic carbon in biomass burning aerosol using optical and chemical measurements" and this is an AGU abstract, not a published paper. Thus, the assumption of 1.1 g/cm3 does not seem justified by the literature references.

Eqn. 5: It would be helpful if the OMF here (and throughout) were labeled as OMF_PM1 to make clear that it is for the bulk PM1 measurement.

P11/L22: it would be helpful to have clarification on what is meant by "organic-salt mixtures" and how this differs from a ZSR model of an organic with a salt.

P11/L32: The compressed film model dynamically partitions material between the bulk and surface dependent on the specified parameters. What does it mean to say that all of the organics were partitioned to the surface here? Was this constrained somehow?

P11/L27: Is this speciation applied within the context of the compressed film model? If so, how were all these different components specified? What was assumed to occur for the organic components that did not partition to the surface? Are they dissolved in the bulk?

P12/L1: How was this decided for the SSA distribution? Where do these parameters come from?

P12/L11: The authors state "Significant correlations were observed between Chl a and total high molecular weight proteins and polyunsaturated fatty acids (R2 = 0.51, p-value

< 0.01)." Can it then be assumed that there is a weak correlation between Chl-a and things that are not mentioned as being correlated, in particular the saturated fatty acids tht the authors note dominate the total?

P12/L19: It seems inconsistent to say that the Chl-a was highest in Bloom 1 with a value of 0.84 while the range is given earlier as up to 1.53.

P13/L5: it is perhaps more appropriate to indicate these as "SSA size distributions produced from natural sea water" rather than as "natural SSA size distributions" given that the reported distributions look quite different than what has been estimated for ambient SSA.

P13/L5: The authors note that the SSA size distributions produced from natural sea water are narrower than those produced from sea salt and suggest this is consistent with addition of surfactant material, citing Fuentes (2013) and Modini (2010). However, they authors might also note that the size distributions from Forestieri (2018) were nominally the same between sea salt and real seawater and from Zabori et al. (2012) were nominally the same for NaCl water and after spiking with succinic acid, although they did observe a notable difference for real Arctic ocean water.

P14/L3: This statement does not seem correct. The size distributions here peak at smaller, or similar modal diameters compared to a number of other studies. For example, the authors compare with Prather et al. (2013). The SSA from sintered glass filters in Prather et al. peak around 80 nm, consistent with the observations here, although the literature distribution is narrower. Hoewver, the SSA distribution produced from wave breaking had a modal diameter much larger. Also, the modal diameters estimated from multi-mode fitting in e.g. Saliba et al. (2019) are much larger in general.

P17/L15: For consistency with the discussion of Ca and Mg EFs, the authors should report the Cl/Na ratios for their lab sea salt experiments in addition to the reported values for seawater.

P19/L14: Where does this factor of 1.5 come from? It seems like it comes from the tartaric acid experiment in Vaattovaara, but it is not clear. Has the variability in this value to different organic species been explored beyond Vaattovaara, in particular saccharides? Is 1.5 reasonable? What is the uncertainty?

P19/L17: What does it mean that the ethanol GFs did not correlate with the other estimates of organic volume fraction? This seems quite important in the context of the OVF interpretation, and is worth some discussion beyond just saying that perhaps the organics that contributed to growth were a subset of those that evaporated. Why would this be the case? Is this consistent with the estimates of the composition?

P19/L23: I am squinting at Table S2 and Fig. 5 and failing to see clearly how the difference in OVF or OMF from deep to mixed layer is statistically significant, or even real. I suggest this be removed unless the authors can justify it further. They note the limited number of samples, but even within these few samples there seems to be sufficient variability to not make this a robust conclusion.

Fig. 9: Which OVF is shown here? The total, I assume.

Fig. 10: Should indicate the particle size.

P24/L11: The word "observed" would better be "required" or "determined." Also, while below the "threshold" indicated to have notable surface tension effects, is this small value reasonable from a physical standpoint? I think that the value they note, < 10 cm3/mol, corresponds to an unreasonably small molecular weight. Additionally, the authors might note that the Forestieri et al. (2018) work focused on CCN while the current work focuses on sub-saturated conditions. As noted above, the hygroscopic response to surface tension depression in different RH regimes can be quite different. In general, greater distinction between sub-saturated and super-saturated measurements is needed throughout the paper.

P24/L16: Which OCEANFILMS model? 1 or 2?

P24/L8: It would be very helpful to the reader if the authors showed the calculated HGF as a function of the assumed critical area and molar volume. As this would likely need to be done for different OVF cases, the authors might consider a low, medium, and high case based on their Fig. 9. It is challenging to see how the compressed film model can resolve the observation-model (ZSR) discrepancy in Fig. 9. If I am understanding what the authors are saying, they are able to do so by tuning of the compressed film parameters. I will be honest and say that from what I understand of the model I don't really believe that the compressed film model can resolve the model-measurement gap here for sub-saturated conditions. There is quite a bit of literature on the relationship between sub- and super-saturated hygroscopicity that the authors might consider (c.f. Wex et al. (2009) and citations therein and that follow). Finally, as noted above already, further details regarding the partitioning of the different components in the context of the compressed film model is needed. What happens to the components that do not go to the surface? What hygroscopicity is assumed?

Fig. 14: I think this should state that this is for the compressed film model where all organics can partition to the surface, not that they are. Their partitioning is dynamic in the model. Same for Fig. 15.

P27/L7: The authors mention this 30 mN/m surface tension depression as being consistent with various SSA proxies, citing Forestieri et al. (2018). However, that paper, as well as Nguyen et al. (2017), show that the fatty acids have negligible impact on activation because the surface tension is dynamic. This aspect, that the surface tension is an evolving property, seems to be lost in the current discussion.

If the DOI for the data set is now known, it should be provided.

Grammar note:

P1/L10: "comprised of" should be "composed of".
* * *
[Figure]

2019.

---

## Referee Comment (RC2) · Christopher Oxford (Referee) · 13 Nov 2019

This document intends to provide information about sea salt aerosol sourced from the southern hemisphere. Justification for the study is given as a lack of southern hemisphere measurements and underestimation of low-level cloud cover. The measurements come from a 23-day ship voyage off the coast of New Zealand. Several chemical speciation measurements were taken along with VH-TDMA (water) and UFO-TDMA (ethanol) measurements. The author uses statistical analysis of the many variables to survey for correlations. Some of those correlations do not have legitimate causation. The document ends by trying to resolve the issues using OCEANFILMS (vs ZSR). The amount of work is significant and clearly represents measurements from the southern hemisphere. Some changes should be made prior to full publication.

Page 3: Line 16: Does the sea salt samples (from manufactured sources) not create salt hydrates?

Page 3: Line 18: OVF not defined.

Page 4: Line 14: HGF not defined

Page 5: Line 8: beta 660 backscatter, pCO2, and DMSsw not previously defined.

Page 5: Line 10: DMS not previously defined

Page 7: Line 4: "Subsequent to heating the SSA was exposed to 90% RH and the hygroscopic growth factor was measured." Please insert a comma or adjust to better display the subordinate clause.

Page 7 Line 4: RH not previously defined.

Page8: DOC not previously defined

Page10: Were the inverted volatility scans used as inputs to the Gysel inversion routine to calculate growth factor as insinuated by equation 4? After performing volatility, the particles shrink some. How is this shrink, prior to hygroscopic growth, represented using the Gysel inversion?

Page 10: Line 14 and 15: recompose sentence to read that sulfate mass was calculated from S, not all inorganics from S measurements.

Page 11: Why would a salt hydrate have a growth factor of 1?

Page 13: I am assuming the number of size distribution modes correlates with the four sintered glass filters. Is this true? If not, please dispel the misconception.

Page 14: Although this may be a little over critical, the natural sea water normalized concentration is missing 1% in Table 1.

Page 14: Line 17 and 18 and Figure 5: how do we know that the non-volatiles (OVFNV) are organic? If you have a proxy for total organic mass and a proxy for semi-volatile

mass, wouldn't the involatile be the difference between the two using assumptions for density?

Page 15: The hygroscopic growth measurements are based on number population (as described in Section 3.4). The volume fraction (used in volatility) is based on both number and diameter. (unless everything is singly charged, the two numbers do not correlate). FYI, 80% of the population is singly charged for this situation. The averaged sampled population from mode 3 is 17% by number and 27% by volume. See table below. These calculations are based on three items: the non-diffusing DMA transfer function (Stolzenburg and McMurry 2008) and your reported DMA 1 settings, the charging fraction as defined by (Wiedensohler 1988), and the reported size distributions in Table 1. In the numbers below, I have multiplied the normalized population numbers in Table 1 by 100,000 for clarity.

In the tables below, columns represent the size distribution modes defined by Table 1. The rows represent the size distributions selected by DMA1 (e.g. +1 is the singly charged size distribution). The percentages are of the total. The first table is by number and the second table is by volume.

Unfortunately, I am unable to place the tables here in this document. Please see supplement for theoretical tables.

Page 16: Feel free to use the numbers from the tables to try to resolve any issues in error in volume fraction. I should note that the numbers above are based on your published average settings and will not be representative of an individual scan.

Page 18-Figure 7 caption: "Stars in bottom right plot represent the mean EF from TEM-EDS measurements of SSA generated from laboratory seawater, dotted error bars show standard deviation in the mean." – I do not see any stars in the panel.

Page 19: OM not previously defined.

Page 19 line 5: tot should be to

Page 19 bottom paragraph: I noticed that the number fractions in the growth factor distribution roughly correlate with charges: the first charge constitutes 80% of the population. How do you know that the lower growth mode isn't the singly charged particles?

Page 20 line 1 through 5: This could be true (using the above tables), but it is likely more complicated. The first size distribution mode could also create the higher growth mode, by theory. I understand that there was statistical correlation, but I find no causal relationship for size distribution 3 being the only size distribution mode related to the second growth factor mode.

Page 20 Figure 8: Is it possible to keep the ordinate of panels (a) and (b) the same to show the increase in HGF due to heating?

Page 28 Line 22 and 23: I do not see any evidence in this work that shows a discrepancy between modeled CCN in the atmosphere and actual CCN measurements during the study. Use of the word "improve" seems inappropriate given the lack of evidence. A verb similar to "change" or "alter" seems more appropriate.

References

Stolzenburg, M. R. and McMurry, P. H. (2008). Equations governing single and tandem DMA configurations and a new lognormal approximation to the transfer function. Aerosol Science and Technology 42:421-432.

Wiedensohler, A. (1988). An approximation of the bipolar charge distribution for particles in the submicron size range. Journal of Aerosol Science 19:387-389.

Please also note the supplement to this comment:
https://www.atmos-chem-phys-discuss.net/acp-2019-797/acp-2019-797-RC2-supplement.pdf

---

## Author Comment (AC2) · 22 Feb 2020

The response to comments from reviewer # 1 has been uploaded as a supplement.

Please also note the supplement to this comment: https://www.atmos-chem-phys-discuss.net/acp-2019-797/acp-2019-797-AC2-supplement.pdf

---

## Author Comment (AC1)

**Reviewer #1 Comments**

**(Authors Response in italics, excerpts from text in bold)**

I have a number of questions and suggestions for clarification, and have some concerns about their interpretation of the sub-saturated hygroscopic growth. This paper should ultimately be publishable, but I think there are a number of aspects that require clarification first.

*The authors thank the reviewer for their detailed review, very helpful comments and broadly positive review.*

*The authors have addressed the concerns surrounding the sub-saturated hygroscopic growth and, in particular, the implementation of the compressed film model. In the previous implementation the use of an unrealistically low molecular volume resulted in a large Raoult term, which was subsequently misattributed to the surface tension effects. The model parameters have been constrained, resulting in a surface tension effect in line with that observed from other sub-saturated water uptake studies.*

*Another change to the manuscript worth noting is the removal of the NaSO4 hydrate component from the computation of fio, which has a subsequent impact on the OVF computed from volatility measurements. This component was removed because it has been shown that the contribution to volatility from NaSO4 hydrates is very small (Rasmussen et al. 2017).*

P1/L9: I suggest "the Aitkin mode" be specified as a size range.

*The authors agree with this statement and the text has been changed.*

> **"…of the particle volume for 50 nm diameter sea spray."**

P1/L16: The particle size should be stated.

*The authors agree with this statement and the text has been changed.*

> P1/L16 **"Nascent 50 nm diameter sea spray aerosol hygroscopic growth…"**

P2/L1: The influence of surface partitioning is quite small on sub-saturated hygroscopic growth. The impact of surface partitioning, and surface tension depression, becomes much more important at very high RH, near 100%. If the authors want to highlight this as a reason for their observations, they need to perform calculations within the main text that illustrate the importance of this effect. Simply stating that surface partitioning can explain the results is insufficient.

The discussion on Page 20, which is in reference to various CCN measurements and not sub-saturated hygroscopic growth measurements, is not directly relevant. There needs to be a clear discussion of the impact at the conditions of the measurements. I suggest this sentence be deleted unless it can be backed up with appropriate calculations.

The details regarding the compressed film model and how it was used (Page 24) are not sufficiently clear to allow a reader to understand what specifically was done.

*As mentioned above interpretation of the surface partitioning was overstated. The calculations have been changed in the manuscript to constrain the model inputs to those that are known to be relevant for SSA. The surface tension effect is reasonably small and doesn't*

*fully describe the observed HGFs. This is now stated in the abstract, discussion and conclusion.*

**Abstract**

**"The compressed film model was used to estimate the influence of surface partitioning and the error in the modelled hygroscopicity was minimised when only the lipid component was partitioned to the surface. The inclusion of surface tension effects somewhat improved the modelled hygroscopicity, however a discrepancy between the observed and modelled hygroscopicity at high organic volume fractions remained."**

*The authors have edited the text to clarify when super saturated conditions are being discussed and added discussion about the influence of surface tension at around 90%RH.*

*Introduction (Page 4 Line 29).* **"…It is worth noting that the impact of surface tension on water uptake in the sub-saturated regime is generally small (Ruehl et al. 2016; Moore et al. 2008)…."**

*The authors have amended the text to explain the application of the compressed film model (in 2.3 Data analysis and in the supplement Page 7 Line 30 and Page 9)*

*e.g. Page 12 Line 14*

**As a counterpoint to the ZSR assumption, which assumes the organic component is dissolved into the bulk, the compressed film model (Ruehl et al., 2016) was applied to explore the influence of partitioning organics to the surface on the nascent SSA water uptake. The composition of the SSA organics are unknown, therefore the compressed film model was computed for organics with a molecular volume of Vorg of 4_10⬜5 m3 mol⬜1 and a molecular area (A0) of 150 square angstroms. The 20 molecular volume was chosen to correspond with the upper limit on the hygroscopicity of organics used in the ZSR assumption (HGF = 1.6). An increase in the compressed film model HGF relative to the ZSR modelled HGF is therefore due to a reduction in surface tension, not to changes in the water activity. The molecular area was chosen to correspond with calculations on sea spray mimics in Forestieri et al. (2018b), who pointed out that to have an impact on surface tension, A0 needs to be in excess of 100 square angstroms.**

P3/L7: I suggest it would be better to refer to the OVF values determined from hygroscopicity measurements as "derived" rather than "observed."

*The authors agree with this statement and the text has been changed.*

**"Exceptions include Quinn et al. (2014) who derived an organic volume fraction…"**

P3/L11: I suggest "volatilizable" is more appropriate than "volatile" to describe the organic material in SSA. As the material exists in the condensed phase, it is not exactly "volatile."

*The authors agree with this statement and the text has been changed.*

**"The organic fraction of SSA appears to be comprised of a volatilisable component which…"**

P3/L13: As the location is given for the Modini paper, it seems appropriate to give it for the Quinn paper as well.

*The authors agree with this statement and the text has been changed.*

> **"An SSA non-volatile organic component has also been observed in the North Pacific and Atlantic Oceans.."**

P3/L15: I do not see how the Modini paper here concluded that there was not a non-volatile organic fraction. Modini et al. characterized only the volatility and hygroscopicity. They did not characterize organic components. Thus, they would not be able to directly address the question of non-volatile, residual organic compounds.

*Modini et al. 2010 didn't suggest the presence of a non-volatile organic fraction. The sentence states that volatility measurements have previously been used to estimate the presence of an organic sea spray fraction.*

P3/L20: As fatty acids tend to have long hydrocarbon tails, their presence is not necessarily consistent with a large hydroxyl fraction, as stated. This is noted in the next sentence. I suggest these are better aligned.

*The authors agree with this statement and the text has been changed to better align the statements.*

> **"..and polysaccharides predisposed to SSA enrichment (O'Dowd, et al. 2015). The composition of SSA organics is characterised by a large fraction.."**

P4/L13: I do not follow how the studies cited in this sentence are "chamber" studies. I don't think any of these are actually chamber studies. Also, suppressed relative to what? Relative to ZSR? This would then conflict with the first part of the sentence. This could be clearer.

*They are studies of artificially generated nascent SSA. Supressed relative to sea salt i.e. consistent with the HGF from ZSR of organics internally mixed with sea salt. The text has been changed to clarify these two points and to improve general clarity.*

> **"Studies using nascent SSA generation chambers have largely indicated that the presence of primary organics suppresses SSA HGFs by 4-17% relative to sea salt (Bates et al., 2012; Fuentes et al., 2011; Modini et al., 2010a; Sellegri et al., 2006; Quinn et al., 2014; Schwier et al., 2015), consistent with the Zdanovskii, Stokes, and Robinson (ZSR) assumption (Stokes and Robinson, 1966)."**

P4/L16: It would be useful if the authors would more clearly distinguish between subsaturated and super-saturated measurements here, as these can be quite different in their response to added organics.

*The text has been changed to point out when CCN measurements are being referred to.*

> *Page 4 Line 21*

> **"Importantly, exceptions have been identified based on CCN measurements which indicate a role of surface tension on SSA water uptake…..The suppression of surface tension has been identified as having a potential impact on the hygroscopicity parameter (kappa) computed from CCN measurements during nascent SSA microcosm experiments. The hygroscopicity parameter was persistently high…."**

P4/L14: Technically, the Ovadnevaite paper is not on SSA. It is on secondary particles in the marine environment. These should be distinguished, as the composition is very different.

*Ovadnevaite 2011 is mentioned on line 16, Ovadnevaite 2017 on line 21. Both are ambient measurements at Mace Head. The focus of the 2011 paper are primary marine organics, and secondary particles are the subject of the 2017 paper. The authors have clarified the text around the reference to the 2017 paper to highlight that the subject of this paper is secondary particles.*

> *Page 4 Line 26*
>
> **"Alternatives to the assumption of full solubility for the organic component of internal carboxylic acid-salt mixtures have been suggested based on laboratory (Ruehl and Wilson, 2014; Ruehl et al., 2016) and field measurements of secondary particles (Ovadnevaite et al., 2017), and applied to SSA analogues (Forestieri et al., 2018)."**

Introduction in General: As the composition of the organic material may (and likely does) vary with particle size, it might be helpful if the authors were to be as explicit as possible in stating the size range of the measurements when they refer to different studies. For example, are the measurements the total submicron? For just smaller particles? The authors might also consider adding further discussion regarding what the literature suggests about variability in the OA chemical composition with size.

*The authors agree. Details have been added throughout the introduction.*

> E.g. Page 3, Introduction, Para 4
>
> **"The organic fraction of sub-200 nm diameter SSA…"**

P6: Particle generation: What concerns do the authors have regarding the representativeness of their size distributions obtained from their particle generation method, and how this might influence particle composition? The observed particle distributions (Fig. 4) differ notably from measurements of sea spray particles from breaking waves (e.g. Prather et al., PNAS, 2013) or estimated from multi-mode fitting of ambient distributions (e.g. Quinn et al., Nat Geosci., 2017; Saliba et al., PNAS, 2019). Differences in size distribution can be indicative of differences in composition. This issue needs to be explicitly discussed, including discussion of potential biases that might result.

*This is a limitation of this study, which is addressed in section 3.2. The authors have added text on P6 that references the discussion in Section 3.2. In addition the text in Section 3.2 has been amended to more clearly reference the potential bias towards higher organic enrichment from using this method.*

> Page 6, Line 2:
>
> **"SSA produced from sintered glass filters does not perfectly represent real world bubble bursting from wave breaking (Collins et al., 2014; Prather et al., 2013) and the limitations to the methods are discussed in further detail in Section 3.2."**
>
> Section 3.2 (Page 15 Line 8)
>
> **"SSA produced from sintered glass filters does not perfectly represent real world bubble bursting from wave breaking (Collins et al., 2014; Prather et al., 2013) but …Observations have shown organic enrichment (King et al., 2013)**

**and also externally mixed organics (Collins et al., 2014) for Aitken and accumulation mode SSA using sintered glass techniques, with slightly higher organic enrichment than that observed using plunging water or wave breaking methods. The use of sintered glass filters may result in primary organics being overrepresented in SSA. Despite the limitations, the use of sintered glass filters allowed an examination of the components of seawater that contribute to SSA organic enrichment."**

Fig. S2: These are not exactly "volatility profiles" as stated in the main text, but instead a comparison of the volatile fraction for sea spray versus sea salt. A volatility profile would be a graph of volatile fraction or fraction remaining versus temperature.

*Description changed in the main text.*

2.2 Measurements/instrumentation, para 4

**"A comparison of the sea salt and sea spray volatility (Fig. S2) was used to calculate the 50 nm SSA organic volume fraction."**

P7/L15: The Vaattovaara et al. (2005) paper indicates that there is negligible growth for 10 nm inorganic particles, but that growth of larger particles in ethanol vapour is not negligible. Furrhter, that paper did not consider sodium chloride. Has it been demonstrated that this method works for particles more representative of those sampled here, that is do sodium chloride (or sea salt) particles not grow? This would be helpful to place uncertainty bounds on the authors measurements.

*The Vaattovaara et al. (2005) include sodium chloride. NaCl (98% purity) is shown it the table 3 and shows no growth in ethanol sub-saturated vapor (86%). That makes also sense with known ethanol solubility information. Furthermore no growth of 100 nm NaCl particles was demonstrated in Joutsensari et al. (2001, ACP).*

P7

**"Growth of sodium chloride and ammonium sulfate in ethanol vapour have been shown to be negligible for preselected diameters up to 100 nm (Vaattovaara et al., 2005; Joutsensaari et al., 2001), while oxidised organics (tartaric, benzoic and citric acid) have growth factors of 1.3 to 1.6 in subsaturated (86%) ethanol vapour."**

Methods: There is a general lack of discussion of uncertainties. Such discussion would be welcome (aka is really needed)

*Further discussion around the uncertainties resulting from the approaches taken has been added. For example text discussing the assumption that only SSA hydrates evaporate at 200 – 400 deg C has been added. Also some text on the potential uncertainty surrounding the computation of fio has been included*

2.3 Data analysis para 3

**"If the SSA samples contained some organics that evaporated between 200-400 _C, these**
**would be incorrectly assigned as inorganic sea salt hydrates, in this respect the computed organic volume fractions could be considered lower limits."**

2.3 Data analysis para 6

**"…Fio was computed as the ratio of natural seawater SSA hydrate volume fraction to laboratory sea salt hydrate volume fraction. These calculations of the sea salt hydrate fraction used the PM1 measurements, but were applied to 50 nm diameter SSA. This is a potential source of uncertainty to the computed OVF, which is sensitive to changes in fio. There would have to be an appreciable difference in the enrichment of a hydrate forming component between the 50 nm SSA and both the PM1 SSA and 50 nm sea salt for this to impact the OVF. Previous observations have shown size dependent enrichment in the sub micron SSA Ca and Mg components for example (Salter et al., 2016; Keene et al., 2007), but this has also been observed for sea salt (Salter et al., 2016). In the context of this study an increase in the volatility due to an increase in hydrates at 50 nm (relative to PM1) is assumed to be reflected in the sea salt volatility and have little impact on the computed OVF."**

P8/L17: The TDMAinv method does not account for multiply charged particles. Are these a concern? Based on the size distributions shown, I would think they would be.

*The doubly charged particles would have a diameter of approximately 75 nm. The HGF difference between 50nm and 75 nm SSA should be reasonably small. With regard to the potential impact on HGF modes, the number fraction of the first HGF mode varies between the seawater samples, ranging from 0.47 to 1, which isn't consistent with a (stable) charge fraction.*

P8/L22: I do not find it clear how the volatility of hydrates is accounted for, nor how consideration of the hydrate proportion yields the organic fraction.

If much of the organic fraction is truly non-volatile, wouldn't this method fail? Or if organic material chars to become non-volatile?

Also, wouldn't this method fail if organics also evaporate between 200C and 400C? Organic volatility tends to be a continuum. Thus, one might not expect a bimodal distribution, as assumed here. How do the authors justify this assumption regarding organic volatility? It may be reasonable, but requires justification.

(Note: if something is non-volatile, then it doesn't evaporate. Thus, the "non-volatile" organics indicated here are not non-volatile, but very low volatility. I strongly suggest the authors adopt a more precise language.)

*The authors agree that the explanation of this approach could have been clearer text has been changed and a schematic with accompanying explanation has been added to the supplement to clarify the approach.*

Section 2.3 para 3.

**"The volume of hydrates is assumed to be a stable proportion of the sea salt volume, and there is assumed to be no contribution to the hydrates from SSA organics. As the organic fraction of internally mixed SSA increases,"**

*The method doesn't fail for large non-volatile OVFs. If the organic fraction is largely non-volatile the slope of Eqn 2 would be lower. If organic material chars to become non-volatile the volume that evaporated at below 200 degrees would be assigned to the SV OVF and the charred residual would be assigned to the low volatility OVF.*

*References cited in the text indicate that the evaporation of organics at 200-400 degrees is minimal. This is based on the observed stepwise SSA volatility – the results in this paper are consistent with this literature. If organics evaporated at 200-400 degC then they would be incorrectly assigned as inorganic sea salt - in this respect the organic fractions are lower limit, and this has been added as a discussion point in the methods. In addition, the observed slopes of the sea spray volatile fraction vs sea salt volatile fraction are less than 1, in particular for the samples with high organic fractions (from the PM1 filter measurements).*

2.3 Data analysis para 3

**"If the SSA samples contained some organics that evaporated between 200-400 _C, these**
**would be incorrectly assigned as inorganic sea salt hydrates, in this respect**
**the computed organic volume fractions could be considered lower limits."**

*The authors agree the use of non-volatile is not strictly accurate and the text throughout has been changed to refer to low volatility organics.*

Eqn. 2 relationship to OVF_tot: I think that these relationships could be stated more clearly. It is not clear, at least to me, why the total OVF would be 1-f (P9/L3). If the OVF_SV = 0, then in Eqn. 2 the value of f is by definition unity. The OVF_tot would then be 1-1 = 0. But, this wouldn't account for evaporation above 400C. So it is unclear to me how this works. It is similarly unclear how the authors end up with OVF_tot values up to 0.9 (Fig. 6) when the max VF values in Fig. S2 reach only 0.23.

Perhaps I am misunderstanding, but I find the discussion here to be unclear, making it difficult to really understand how the OVF_tot values were determined. I suggest revision is needed. I strongly encourage the authors to include graphs of VF (or VFR) versus temperature so that the reader can clearly see what must be a step change after _400C indicative of evaporation of "non-volatile" components. There must be huge differences between the samples with OVF_tot = 0.9 and those with small values. It would be useful to the reader to see these.

*The text has been changed as described in the previous comment and a schematic with accompanying explanation has been added to the supplement to clarify the approach.*

*If OVF_SV =0, f does not have to be unity. There could definitely be an intercept of 0 and a slope of < 1. This would indicate a low volatility organic fraction, but no sv organic fraction.*

*The OVF is driven by f, the slope of Eqn 2 over 200 – 400 degrees.*

*A plot of the volatility profiles for the natural seawater SSA has been included in Fig. S3.*

P15/L5: The authors need to provide much more detail here regarding how they apportion things to the different modes. There must ultimately be full consistency with all the measurements. How do they decide how much of the volatile and "non-volatile" material goes between the three modes that overlap? Does this add up appropriately to, hypothetically, reproduce the observations? It is also not clear whether the authors assumptions allow for any salts in modes 1-3, or whether these are limited to mode 4. As written, they only indicate salts in mode 4. If this is the case, then the assumptions there are inconsistent with the VF observations. There must be a salt component at 50 nm, based on their interpretation, and thus there must be some salt in either Mode 1, 2, or 3. Additionally, the authors point the reader to Section 3.4 to justify their split, but it is not evident after reading Section 3.4 exactly how they made this determination. They need to be more explicit here and (i) fully justify their choices while (ii) demonstrating the internal consistency. As best I can tell from the range of sizes considered, the authors do not have an independent

constraint on the composition of Mode 4 since it contributes negligibly to the number concentration below 150 nm, although will have a large influence on the overall mass.

*The authors agree with these concerns and on reviewing this section don't think there is the sufficient evidence to apportion the organic fraction to the lognormal modes. The externally mixed HGFs are present for both sea salt and natural sea spray particles – therefore does not provide necessary information on the organic composition. As a result the comparison between VH-TDMA derive PM! OMF and filter measured PM1 OMF has been removed. A comparison between the 50 nm OVF and PM1 OMF remains (Figure 6).*

Section 3.3 SSA composition

**"The partitioning of organics to the lognormal modes has been removed. The correlation between the 50 nm OVF and the PM1 OMF are still presented in Figure 6."**

OVF correlations: What does the slope of OVF versus some seawater metric (e.g.concentration of alkanes) mean? These are reported, but the meaning is not clear as the OVF is a fraction of the total PM. Also, shouldn't these slopes have units?

*The slope provides a rough comparison of the propensity of the seawater species for being enriched into SSA, on average. For example, per unit of seawater high molecular weight carbohydrates there is a larger increase in the $OVF_{sv}$ (slope $10^{-3}$), than there is per unit of seawater high molecular weight proteins (slope $10^{-4}$). Units have been added to the text.*

P15/L10: An assumption of an organic density of 1.1 g/cm3 seems at odds with the determination that species such as saccharides dominated the composition (based on the large hydroxyl fraction). Can this be further justified? Citation of Modini is insufficient, as that paper simply assumed 1.1 g/cm3 based on Keene et al. (2007) and thus is not an independent determination. Then, Keene et al. (2007) do not actually determine this, but state it is estimated based on Schkolnik et al. (2006). The title of Schkolnik et al. (2006) is "Constraining the density and complex refractive index of elemental and organic carbon in biomass burning aerosol using optical and chemical measurements" and this is an AGU abstract, not a published paper. Thus, the assumption of 1.1 g/cm3 does not seem justified by the literature references.

*This density is no longer required because the apportionment of organics to the lognormal modes and the calculation of PM1 OMF from TDMA data has been removed.*

*In general though, the authors agree and in the computation of the organic volume fractions for implementation in the compressed film model the applied densities were from Petters et al. 2009 (Lipids and Polysaccharides) and Mikhalov et al. 2004 (Proteins) have been used (See supplement Water Uptake).*

Eqn. 5: It would be helpful if the OMF here (and throughout) were labelled as OMF_PM1 to make clear that it is for the bulk PM1 measurement.

*The authors agree with this comment and reference to PM1 OMF has been added throughout manuscript.*

P11/L22: it would be helpful to have clarification on what is meant by "organic-salt mixtures" and how this differs from a ZSR model of an organic with a salt.

*The authors agree this is not clear and the text has been changed.*

2.3 Data analysis para 12

**"…as an upper limit for organics that could possibly be present in SSA"**

P11/L32: The compressed film model dynamically partitions material between the bulk and surface dependent on the specified parameters. What does it mean to say that all of the organics were partitioned to the surface here? Was this constrained somehow?

*The text has subsequently been changed. However, this the authors agree the wording should be that all of the organics were able/available to be portioned to the surface, the actual partitioning is dynamic as the reviewer pointed out. This has been reflected throughout.*

2.3 Data analysis 2nd last para

**"HGFs were computed using the compressed film model for three cases, assuming that just the lipids are able to partition to the surface, that the lipids and the polysaccharides are able to partition to the surface and assuming that all of the organics are able partition to the surface."**

P11/L27: Is this speciation applied within the context of the compressed film model? If so, how were all these different components specified? What was assumed to occur for the organic components that did not partition to the surface? Are they dissolved in the bulk?

*That is correct, these components were speciated for the compressed film model, the text has now changed to make this more explicit. Yes, the components not partitioned to the surface were dissolved into the bulk. Further information was added to Supplement regarding the implementation of the compressed film model.*

2.3 Data analysis 2nd last para

**"The speciation of organics into molecular classes was calculated from the functional group concentrations as shown in Burrows et al. (2014) and applied in the compressed film model. HGFs were computed using the compressed film model for three cases, assuming that just the lipids are able to partition to the surface, that the lipids and the polysaccharides are able to partition to the surface and assuming that all of the organics are able partition to the surface."**

P12/L1: How was this decided for the SSA distribution? Where do these parameters come from?

*This is an example case based on the parameters used to fit SSA distributions from Modini et al. 2015 and Quinn et al. 2017. References to Modini et al. 2015 and Quinn et al. 2017 have been added to the text .*

P12/L11: The authors state "Significant correlations were observed between Chl a and total high molecular weight proteins and polyunsaturated fatty acids (R2 = 0.51, p-value < 0.01)." Can it then be assumed that there is a weak correlation between Chl-a and things that are not mentioned as being correlated, in particular the saturated fatty acids that the authors note dominate the total?

*There was very poor correlation for high molecular weight carbohydrates, alkanes, saturated FAs and monounsaturated FAs (p-vals > 0.15, R2 < 0.1).*

P12/L19: It seems inconsistent to say that the Chl-a was highest in Bloom 1 with a value of 0.84 while the range is given earlier as up to 1.53.

*Chl-a ranged from 0.3 to 1.53 during bloom 1 and averaged 0.84. Text changed for clarity.*

> **"…and displayed the highest average Chl -a concentrations of 0.84."**

P13/L5: it is perhaps more appropriate to indicate these as "SSA size distributions produced from natural sea water" rather than as "natural SSA size distributions" given that the reported distributions look quite different than what has been estimated for ambient SSA.

*The authors agree with this comment and the text has been changed.*

> **"Size distributions generated from natural seawater were slightly…"**

P13/L5: The authors note that the SSA size distributions produced from natural sea water are narrower than those produced from sea salt and suggest this is consistent with addition of surfactant material, citing Fuentes (2013) and Modini (2010). However, they authors might also note that the size distributions from Forestieri (2018) were nominally the same between sea salt and real seawater and from Zabori et al. (2012) were nominally the same for NaCl water and after spiking with succinic acid, although they did observe a notable difference for real Arctic ocean water.

*The authors agree with this comment and the text has been changed.*

> **"Differences between the shape of inorganic sea salt and organically enriched sea spray size distributions has not been observed in all studies (Forestieri et al. 2018, Zabori et al. 2012)."**

P14/L3: This statement does not seem correct. The size distributions here peak at smaller, or similar modal diameters compared to a number of other studies. For example, the authors compare with Prather et al. (2013). The SSA from sintered glass filters in Prather et al. peak around 80 nm, consistent with the observations here, although the literature distribution is narrower. Hoewver, the SSA distribution produced from wave breaking had a modal diameter much larger. Also, the modal diameters estimated from multi-mode fitting in e.g. Saliba et al. (2019) are much larger in general.

*This statement should reference studies using the same generation method. Text has been changed to make this clearer.*

> 3.2 *SSA size distributions* Para 2

> **"The shape of the nascent SSA size distribution was broadly similar to nascent SSA size distributions observed in previous studies which also used sintered glass filters, but shifted to slightly larger mean diameters."**

P17/L15: For consistency with the discussion of Ca and Mg EFs, the authors should report the Cl/Na ratios for their lab sea salt experiments in addition to the reported values for seawater.

*The authors agree and this has been added*

> **"It is also worth noting that the mass ratio of Cl□ to Na+ from sea salt TEM-EDS measurements was much lower than that for seawater, 1.3 \pm 1, however the uncertainty was very large."**

P19/L14: Where does this factor of 1.5 come from? It seems like it comes from the tartaric acid experiment in Vaattovaara, but it is not clear. Has the variability in this value to different

organic species been explored beyond Vaattovaara, in particular saccharides? Is 1.5 reasonable? What is the uncertainty?

*Correct, the HGF comes from Vaattovaara et al. 2005. No, the variability hasn't been explored for those common marine organic species. Text has been added to acknowledge the limitation of this technique.*

> **"Applying the ZSR assumption with an organic growth factor of 1.5 and a sea salt growth factor of 1, based on UFO-TDMA measurements of oxidised organics (tartaric, benzoic and citric acid) and sodium chloride, respectively (Vaattovaara et al., 2005; Joutsensaari et al., 2001). The measured ethanol growth factors correspond to moderately oxidised organic volume fractions averaging 35 _ 5%, when the two component ZSR model above is applied. The ethanol growth factor for species commonly observed in SSA, such as polysaccharides, proteins and lipids is unknown, and therefore the representativeness of the ZSR model for primary marine aerosol is highly uncertain."**

P19/L17: What does it mean that the ethanol GFs did not correlate with the other estimates of organic volume fraction? This seems quite important in the context of the OVF interpretation, and is worth some discussion beyond just saying that perhaps the organics that contributed to growth were a subset of those that evaporated. Why would this be the case? Is this consistent with the estimates of the composition?

*Text has been added to acknowledge the limitation of this technique.*

> **"The variability due to SSA diameter in the ethanol growth factors measured at 15 to 50 nm were all within experimental error once a correction for the Kelvin effect was applied. There were no significant correlations with the 50 nm ethanol growth factor and the organic volume fraction calculated from volatility and PM1 organic mass fractions. The species responsible for the observed ethanol growth can't be determined without further reference measurements for sea spray. The ethanol growth factor of volatilised SSA (for sample U7520) was 1:03_0:03 at 200 _C, and averaged 1:01_0:03 between 250 and 400 _C, suggesting that the component contributing to ethanol growth was largely semi-volatile. The component that contributed to ethanol growth was more constant than the OVFSV measured using the VH-TDMA, suggesting that it could have been a subset
> of the total volatile organic component."**

P19/L23: I am squinting at Table S2 and Fig. 5 and failing to see clearly how the difference in OVF or OMF from deep to mixed layer is statistically significant, or even real. I suggest this be removed unless the authors can justify it further. They note the limited number of samples, but even within these few samples there seems to be sufficient variability to not make this a robust conclusion.

*The authors agree the variability is large, and the effect is marginal. The paragraph has been removed.*

Fig. 9: Which OVF is shown here? The total, I assume.

*Yes total OVF, this has been added to the caption.*

Fig. 10: Should indicate the particle size.

The authors agree and "50 nm" has been added to the figure caption

P24/L11: The word "observed" would better be "required" or "determined." Also, while below the "threshold" indicated to have notable surface tension effects, is this small value reasonable from a physical standpoint? I think that the value they note, < 10 cm3/mol, corresponds to an unreasonably small molecular weight. Additionally, the authors might note that the Forestieri et al. (2018) work focused on CCN while the current work focuses on sub-saturated conditions. As noted above, the hygroscopic response to surface tension depression in different RH regimes can be quite different. In general, greater distinction between sub-saturated and super-saturated measurements is needed throughout the paper.

*This approach has been revised and this text removed – the Vorg that minimised the error in the existing approach was unreasonably small (as described in previous comments). Instead the authors focus on the surface tension effects for model parameters known to be relevant for these species. Throughout the manuscript the text has been clarified to point out the whether sub or super-saturated regimes are being considered, and to point out that the literature shows a small effect of surface tension for typical sub-saturated measurements (~90%RH).*

P24/L16: Which OCEANFILMS model? 1 or 2?

*OCEANFILMS-2, the text has been changed to clarify this point.*

P24/L8: It would be very helpful to the reader if the authors showed the calculated HGF as a function of the assumed critical area and molar volume. As this would likely need to be done for different OVF cases, the authors might consider a low, medium, and high case based on their Fig. 9. It is challenging to see how the compressed film model can resolve the observation-model (ZSR) discrepancy in Fig. 9. If I am understanding what the authors are saying, they are able to do so by tuning of the compressed film parameters. I will be honest and say that from what I understand of the model I don't really believe that the compressed film model can resolve the model-measurement gap here for sub-saturated conditions. There is quite a bit of literature on the relationship between sub- and super-saturated hygroscopicity that the authors might consider (c.f. Wex et al. (2009) and citations therein and that follow).

Finally, as noted above already, further details regarding the partitioning of the different components in the context of the compressed film model is needed. What happens to the components that do not go to the surface? What hygroscopicity is assumed?

*The compressed film model approach has been revised to focus on the surface tension effects for model parameters known to be relevant for these species. The surface tension effects presented in the revised version are reasonably modest and in line with those references the reviewer pointed out.*

*Only one set of compressed film model parameters have been applied in the revised approach, therefore a plot of these parameters vs the computed HGF has not been included. Vorg may have a large effect on the Raoult term in the compressed film model – but this isn't particularly relevant.*

*The discussion around the results from the compressed film model has been changed and a plot has been added in the supplement showing the showing the compressed film model calculated surface tension as a function of OVF. Further detail has been added on the implementation of the compressed film model, in particular the supplement.*

Section 4.2 SSA water uptake, para 4:

**"Directly comparing the HGFs modelled using the compressed film model and those modelled using the ZSR assumption, as in Fig. 14, highlights the contribution of surface tension to the observed SSA HGF. The surface tension effects on HGF observed at high organic volume fractions is up to 0.05, however this does not account for the reduction in HGF predicted by the ZSR assumption, i.e. by Raoult's Law. The modest impact of decreased surface tension of HGF is consistent with previous studies on sub-saturated water uptake (Ruehl et al., 2016; Moore et al., 2008). Despite the inclusion of the surface tension effect there was still a significant discrepancy between the observed and modelled HGFs, even when the relatively large uncertainty is the OVF is considered. "**

Fig. 14: I think this should state that this is for the compressed film model where all organics can partition to the surface, not that they are. Their partitioning is dynamic in the model. Same for Fig. 15.

*The authors agree, the figure has been changed but the updated caption reflects this comment.*

Caption

**"Compressed film model output is for case where lipid fraction can partition to the particle surface."**

P27/L7: The authors mention this 30 mN/m surface tension depression as being consistent with various SSA proxies, citing Forestieri et al. (2018). However, that paper, as well as Nguyen et al. (2017), show that the fatty acids have negligible impact on activation because the surface tension is dynamic. This aspect, that the surface tension is an evolving property, seems to be lost in the current discussion.

*This discussion of the potential for surface tension to impact CCN concentrations has been removed. This section has been updated to discuss the potential difference between the CCN computed using ZSR modelled and measured HGFs.*

If the DOI for the data set is now known, it should be provided.

*Not yet available, there has been some delays in deciding how SOAP data (more broadly) will be deposited, but will be provided as soon as it is available.*

P1/L10: "comprised of" should be "composed of".

*The authors agree and the text has been changed.*

**Reviewer #2 Comments**

**(Authors Response in italics, excerpts from text in bold)**

This document intends to provide information about sea salt aerosol sourced from the southern hemisphere. Justification for the study is given as a lack of southern hemisphere measurements and underestimation of low-level cloud cover. The measurements come from a 23-day ship voyage off the coast of New Zealand. Several chemical speciation measurements were taken along with VH-TDMA (water) and UFO-TDMA (ethanol) measurements. The author uses statistical analysis of the many variables to survey for correlations. Some of those correlations do not have legitimate causation. The document ends by trying to resolve the issues using OCEANFILMS (vs ZSR). The amount of work is significant and clearly represents measurements from the southern hemisphere. Some changes should be made prior to full publication.

*The authors thank the reviewer for their detailed review, helpful comments/suggestions and broadly positive review.*

*The authors have addressed concerns surrounding causation between lognormal modes and composition, for example the apportionment of organics to lognormal modes has since been removed in response to the reviewers comments.*

*Another change to the manuscript worth noting is the removal of the NaSO4 hydrate component from the computation of fio, which has a subsequent impact on the OVF computed from volatility measurements. This component was removed because it has been shown that the contribution to volatility from NaSO4 hydrates very small (Rasmussen et al. 2017).*

Page 3: Line 16: Does the sea salt samples (from manufactured sources) not create salt hydrates?

*Yes the laboratory sea salt and the inorganic sea salt component of sea spray both contain hydrates and this needs to be accounted for.*

Page 3: Line 18: OVF not defined.

*Defined earlier "Exceptions include Quinn et al. (2014), who observed an organic volume fraction (OVF) of up to 0.8 using CCN measurements"*

Page 4: Line 14: HGF not defined

*The text has been changed and the first use of HGF is now defined.*

Page 5: Line 8: beta 660 backscatter, pCO2, and DMSsw not previously defined.

*The text has been simplified.*

Section 2.1 SOAP voyage

**"seawater parameters (Chl-a, dimethyl sulfide, and carbon dioxide concentrations)"**

Page 5: Line 10: DMS not previously defined

*The text has been changed and the first use of DMS is now defined.*

Page 7: Line 4: "Subsequent to heating the SSA was exposed to 90% RH and the hygroscopic growth factor was measured." Please insert a comma or adjust to better display the subordinate clause.

*The authors agree and the text has been simplified.*

**"After heating the SSA hygroscopic growth factor at 90% RH was measured."**

Page 7 Line 4: RH not previously defined.

*The authors agree and the first use of RH has been defined.*

Page8: DOC not previously defined

*The authors agree and the first use of DOC has been defined.*

Page10: Were the inverted volatility scans used as inputs to the Gysel inversion routine to calculate growth factor as insinuated by equation 4? After performing volatility, the particles shrink some. How is this shrink, prior to hygroscopic growth, represented using the Gysel inversion?

*Volatility measurements and hygroscopicity measurements were inverted separately. A single VGF mode was used to correct HGF for volatility. A comment on this has been added*

*2.3 Data analysis.*

**"Volatility measurements and hygroscopicity measurements were inverted separately and a single VGF mode was used to correct HGF for volatility."**

Page 10: Line 14 and 15: recompose sentence to read that sulfate mass was calculated from S, not all inorganics from S measurements.

The authors agree and the sentence has been changed.

Page 11, Line 8
**"The inorganic mass (IM) was computed as the sum of Na, Mg, SO4, Cl, K, Ca, Zn, Br and Sr. The measured S mass was used to calculate the SO4 mass, all S was assumed to be in the form of SO4."**

Page 11: Why would a salt hydrate have a growth factor of 1?

*The hydrate, water component has a GF of 1. The anhydrous salt has a higher HGF than the hydrated salt. This was a way of explicitly including the hydrate volume fraction (which varies between samples) in the ZSR assumption.*

Page 13: I am assuming the number of size distribution modes correlates with the four sintered glass filters. Is this true? If not, please dispel the misconception.

*No not necessarily – for example Fuentes et al. 2010 used a single porosity sintered glass frit and fitted 4 lognormal modes. This has been pointed out in the text.*

*Page 14 Line 2*

***"The measured size distributions were broken up into four log-normal modes characterised by geometric mean diameters ranging from 33 to 320 nm, as seen in Fig. 4. This is consistent with the number of lognormal modes fitted by***

***Fuentes et al. 2010** **and is not a direct result of the use of multiple glass filters in this study."***

Page 14: Although this may be a little over critical, the natural sea water normalized concentration is missing 1% in Table 1.

*A rounding issue, an extra decimal point has been included for clarity.*

Page 14: Line 17 and 18 and Figure 5: how do we know that the non-volatiles (OVFNV) are organic? If you have a proxy for total organic mass and a proxy for semi-volatile mass, wouldn't the involatile be the difference between the two using assumptions for density?

*The linear model provides a calculated total organic volume fraction and semi-volatile organic volume fraction (for preselected 50 nm SSA). The non-volatility/low volatility component is the difference between these two values.*

Page 15: The hygroscopic growth measurements are based on number population (as described in Section 3.4). The volume fraction (used in volatility) is based on both number and diameter. (unless everything is singly charged, the two numbers do not correlate). FYI, 80% of the population is singly charged for this situation. The averaged sampled population from mode 3 is 17% by number and 27% by volume. See table below. These calculations are based on three items: the non-diffusing DMA transfer function (

Stolzenburg and McMurry 2008) and your reported DMA 1 settings, the charging fraction as defined by (Wiedensohler 1988), and the reported size distributions in Table 1. In the numbers below, I have multiplied the normalized population numbers in Table 1 by 100,000 for clarity.

Page 16: Feel free to use the numbers above to try to resolve any issues in error in volume fraction. I should note that the numbers above are based on your published average settings and will not be representative of an individual scan.

*Thank you kindly for your helpful information.*

*This apportionment of the organic fraction (based on volatility) to the lognormal modes has since been removed in the absence of any size resolved composition measurements (particularly around the accumulation mode), in response to questions from reviewer #1.*

Page 18-Figure 7 caption: "Stars in bottom right plot represent the mean EF from TEM-EDS measurements of SSA generated from laboratory seawater, dotted error bars show standard deviation in the mean." – I do not see any stars in the panel.

*"Stars" changed to "Triangles" to reflect the figure.*

Page 19: OM not previously defined.

*First use of OM now defined in the text*

Page 19 line 5: tot should be to

*Text changed to fix typo*

Page 19 bottom paragraph: I noticed that the number fractions in the growth factor distribution roughly correlate with charges: the first charge constitutes 80% of the population. How do you know that the lower growth mode isn't the singly charged particles?

*The number fraction of the first HGF mode varies between the seawater samples, ranging from 0.47 to 1, which isn't consistent with a (stable) charge fraction.*

*The doubly charged particles would have a diameter of approximately 75 nm. The HGF difference between 50nm and 75 nm SSA should be reasonably small.*

Page 20 line 1 through 5: This could be true (using the above tables), but it is likely more complicated. The first size distribution mode could also create the higher growth mode, by theory. I understand that there was statistical correlation, but I find no causal relationship for size distribution 3 being the only size distribution mode related to the second growth factor mode.

*The authors agree that the relationship between the lognormal modes and HGF modes/composition is not certain. The text has been amended to reference the possibility of different lognormal mode compositions, but acknowledging the uncertainty/limitations in this study.*

> *Page 21, Line 7*
>
> **"The fraction of the second HGF mode at 50 nm correlated with the proportion of lognormal mode 3 (R2 of 0.39, p-value < 0.01, and slope of 0:87_0:3). This suggests that the lognormal modes may have different composition and/or morphology, which has previously been observed for nascent SSA (Collins et al., 2013), however in the absence of size resolved compositional measurements further conclusions are not possible."**

Page 20 Figure 8: Is it possible to keep the ordinate of panels (a) and (b) the same to show the increase in HGF due to heating?

*Yes, the authors agree that this would be clearer and the figure has been changed to make the y-axis scales aligned.*

Page 28 Line 22 and 23: I do not see any evidence in this work that shows a discrepancy between modeled CCN in the atmosphere and actual CCN measurements during the study. Use of the word "improve" seems inappropriate given the lack of evidence. A verb similar to "change" or "alter" seems more appropriate.

*This text has subsequently been changed in response to comments from reviewer #1 and no longer uses this wording.*

---

## Author Response (AR3)

**Reviewer comments, 2[nd] review.**

**Sea spray aerosol organic enrichment, water uptake and surface tension effects**

Luke T. Cravigan[1], Marc D. Mallet[1,a], Petri Vaattovaara[2], Mike J. Harvey[3], Cliff S. Law[3,4], Robin L. Modini[5,b], Lynn M. Russell[5], Ed Stelcer[6,c], David D. Cohen[6], Greg Olsen[7], Karl Safi[7], Timothy J. Burrell[3], and Zoran Ristovski[1]

**(Authors Response in italics, excerpts from text in bold)**

The authors have made notable revisions to their manuscript, and I find it most certainly improved. I do, however, have remaining questions that derive from the revisions that I think should be addressed prior to publication.

*The authors thank the reviewer for their detailed review and consideration of the revised manuscript and very helpful comments.*

The caption for Fig. 9 seems incorrect. I think that the blue and green are switched, as a ZSR mixing model would lead to a lower calculated GF at a given OVF when the organics are assumed to have a GF = 1 compared to a GF = 1.6.

> *The authors agree with this statement and the caption has been changed.*

> **HGF modelled using ZSR assumption shown assuming an organic HGF of 1 (green) and an organic HGF of 1.6 (blue) shown**

When the authors now state "An increase in the compressed film model HGF relative to the ZSR modelled HGF is therefore due to a reduction in surface tension, not to changes in the water activity," I'm not sure this is correct. The activity is changing because material is partitioning to the surface and therefore no longer contributing to solubility. So, the reduction in surface tension here offsets the change in water activity, I think.

> *The authors agree with the reviewer, however this sentence has been removed upon review of the compressed film model results (see comments responses below).*

P1/L9: For clarity, would be good to say "The composition of the submicron organic fraction" rather than just "the organic fraction" as the previous sentence breaks things into size ranges.
> *The authors agree with this statement and the text has been changed.*

> **P1/L9**

> **"The composition of the submicron organic fraction was consistent throughout the voyage and was largely composed of a…."**

P3/L2: I suggest adding "dependent on the method used and size range investigated."

> *The authors agree with the suggestion and the text has been changed.*

> **P2/L32**

**"Chamber observations of nascent SSA universally indicate the presence of an internally mixed organic component, with the organic contribution varying between studies from approximately 4% - 80% by volume, dependent on the method used and size range investigated.."**

P3/L5: I suggest changing to "based on water uptake methods, where volume mixing rules are assumed,

*The authors agree with the suggestion and the text has been changed.*

**P3/L3**
**"Estimates of the sub-100 nm SSA organic fraction based on water uptake methods, where volume mixing rules are assumed, are of the order of 5 - 37%.."**

P3/L12: I suggest it be clarified that the volatilizable component is not the only type of organic present.

*The authors agree with the suggestion and the text has been changed.*

**P3/L11**
**"The organic fraction of sub-200 nm diameter SSA appears to be comprised of a volatilisable component which evaporates at approximately 150 - 200°C and comprises of the…"**

P4/L20: The authors updated this sentence, but it is now unclear to me how this is consistent with ZSR. Consistent in what manner? Above, the authors noted that the organic fraction in small SSA can be much greater than 4-17%. Also, in the list of references given few have explicitly shown that ZSR works by making independent measurements of composition and hygroscopicity, and then comparing. I suggest this be clarified.

*The authors agree and the text has been changed.*

**"Studies using nascent SSA generation chambers have largely indicated that the presence of primary organics suppresses sub-200 nm diameter SSA hygroscopic growth factors (HGFs) by 4-17% relative to sea salt (Bates et al., 2012; Fuentes et al., 2011; Modini et al., 2010a). Simultaneous measurement of SSA composition and supersaturated water uptake have indicated that the observed hygroscopicity is consistent with the Zdanovskii, Stokes, and Robinson (ZSR) assumption (Stokes and Robinson, 1966; Schwier et al., 2017)."**

Regarding the organic volume fraction derived by volatility, the authors note in their response that if material charred it would be assigned to the low volatility organic fraction. However, this assumes that the charred material would evaporate at all. This is not guaranteed. In OC/EC analysis, charred organics typically do not evaporate until very high temperatures. These organics would be mis-attributed to salt. This aspect could be clarified further.

*The authors don't think that charred material would be classified as salt. Salt is classified by the volatility at 200 - 400 deg C, the more volatility over this temperature range the more hydrates and therefore the more sea salt (as a fraction of volume). The assumption is that charred material would reduce the volatility at 200-400 degC and therefore be classified as low volatility organics.*

*Mention of the possible misattribution of charred organics has been added to the methods section.*

**P9/L17 " It is also possible that the charring of semi volatile organics could also lead to mis-attribution as low volatility organics. "**

P11/L5: It would be good if the authors clarified here that this is now for PM1, as the discussion to this point about volatility is for small particles.

*The authors agree with the statement and the text has been changed.*

**P11/L5 "The PM1 organic mass fraction from SSA samples collected on filters was…"**

P14/L4: Certainly the size distributions obtained are a result of the glass filters used. It is established that use of different generation methods leads to different size distributions. Perhaps the authors' point here is that the particulars of these distributions are not a result of using four filters, but of using filters in general. Also, this seems to conflict with the authors statement on P15/L9 that the use of four filters is key to getting a broader distribution compared to studies that use different filter configurations. I suggest that this could be clarified.

*The size distributions from the glass filter is generally similar in shape - and represented by 4 lognormal modes. Using 4 sintered glass filters broadens the distribution by spreading the contribution more evenly among the lognormal modes. The text has been changed to clarify these points.*

**P13/L32**
**"This is consistent with the number of lognormal modes fitted by Fuentes et al. (2010) and is not a direct result of the use of four glass filters in this study, but a result of using sintered glass filters in general."**

**P15/L2**
**"SSA produced from sintered glass filters does not perfectly represent real world bubble bursting from wave breaking (Collins et al., 2014; Prather et al., 2013) but the use of four glass filters with different pore sizes resulted in a broader distribution than other measurements of nascent 5 SSA using glass filters (Collins et al., 2014; Fuentes et al., 2010; Keene et al., 2007; Mallet et al., 2016). For example, the normalised number contributions from mode 1 and mode 4 was three times greater than that observed by Fuentes et al. (2010)."**

P15/L20: I'll encourage the authors to modify this to say "(generally greater than 0.4 and as high as 0.85)"

*The authors agree and the text has been changed*

**P16/L5**
**"The 50 nm OVF was highest during bloom 1 (generally greater than 0.4 and as high as 0.85), which..."**

Fig. 6: There are a few errors in the revised Fig. 6. The slope of the blue lines is definitely not 1.3. It is much larger. I think this was mislabeled. Also, I highly doubt the R2 for the blue points is 0.68. I think it is closer to 0.35. I think this higher R2 reported is only obtained when

the volatility-derived OVF values near zero when the filter measurements are > 0 are excluded. Similarly, I do not think the R2 value for the red points is 0.27 based on the measurements shown; it is likely lower. I'll note that these near-zero points were not included in the previous version of Fig. 6. It appears that these near-zero points had non-zero values in the previous version. Have these been revised? Or is this a typo? It would be useful if it were clarified where these additional measurements came from. Related, all R2 values discussed on P17 and P18 should be revised accordingly if these near-zero points are correct.

*An outdated version of the organic mass fraction (filter vs TDMA) was inadvertently added. The figure has been corrected, and the in text R² values have been double checked.*

[Figure]

**Figure 6.** Comparison of 50 nm organic volume fraction calculated from volatility measurements (using VH-TDMA) and the $PM_1$ OMF measured using FTIR/IBA on filter samples. Volatile fractions shown in red, total organic fractions shown in green.

P17/L8: I would argue that if the volatilities differ by hundreds of degrees (as is true for the semi-volatile and low-volatilty components) their composition is necessarily quite different. Perhaps the authors could clarify here that they seem to mean the fractional contributions of the different organic components were similar, which is not the same as saying that the composition is similar.

*Agree- text has been changed*

**P17/L2**
**"The correlations suggest that the fractional contributions to the volatile and low volatility OVFs from the different seawater organic components measured in this study were similar, but the semi-volatile OVF displayed a higher contribution from aliphatic, lipid like species"**

P19/L11: It would still be good to indicate this is for PM1.

*The authors agree and the text has been changed*

**P17/L8**
**"Alcohol functional groups contributed 77 ± 8 % of the PM1 SSA organic mass (OM), alkanes...."**

P19/L31: The authors now note explicitly that the estimated OVF from the UFO-TDMA measurements do not correlate with either the semi-volatile or low-volatility OVF values, which is an interesting addition. Yet, they note that the UFO-derived OVF is likely associated with the semi-volatile components, and further that it must be a subset of the semi volatile components. If this is the case, then I would think that the UFO-derived OVF values should be universally lower than the semi-volatile OVF values. However, the UFO derived OVF values are larger (35%) than the semi-volatile OVF values (max ~15%). Can this be explained.

*Further detail has been added regarding the limitation of the application of UFO-TDMA GFs to SSA and some explanation for the potential over-prediction of OVF from the UFO-TDMA.*

**"Ethanol growth factors measured using the UFO-TDMA for preselected 50 nm diameter SSA were 1.22 pm 0.02 (mean pm sd) and were largely invariable for all of the water samples examined. The ZSR assumption was used with an organic growth factor of 1.5 and a sea salt growth factor of 1, based on UFO-TDMA measurements of oxidised organics (tartaric, benzoic and citric acid) and sodium chloride, respectively (Vaattovaara et al., 2005; Joutsensaari et al., 2001). The measured ethanol growth factors correspond to moderately oxidised organic volume fractions averaging 35 _ 5%, when the two component ZSR model above is applied. The ethanol growth factor for species commonly observed in SSA, such as polysaccharides, proteins and lipids are not known, and therefore the representativeness of the ZSR model for primary marine aerosol is highly uncertain. For example some of the fatty acids and alkanes with the highest concentrations in the seawater are soluble (Myristic and Palmitic acid, Eicosane and Docosane) or very soluble (Lauric acid) in ethanol and could potentially display high ethanol growth factors (Haynes, 2018). Chlorophyll-a is also very soluble in ethanol (Haynes, 2018). If the ethanol growth factor of the SSA organic component is larger than 1.5, as assumed in the ZSR assumption, then the computed OVF will be overestimated. The variability due to SSA diameter in the ethanol growth factors measured at 15 to 50 nm were all within experimental error once a correction for the Kelvin effect was applied. There were no significant correlations with the 50 nm ethanol growth factor and the organic volume fraction calculated from volatility and PM1 organic mass fractions. The species responsible for the observed ethanol growth can't be determined without further reference measurements for sea spray. The ethanol growth factor of volatilised SSA (for sample U7520) was 1.03 pm 0.03 at 200 deg C, and averaged 1.01 pm 0.03 between 250 and 400 deg C, suggesting that the component contributing to ethanol growth was largely semi-volatile. The component that contributed to ethanol growth was more constant than the OVFSV measured using the VH-TDMA, suggesting that it could have been a subset of the total volatile organic component. The high UFO-TDMA OVF relative to the VH-TDMA OVF could suggest that the computed OVF from the UFO-TDMA is overestimated."**

Fig. 9: The x-axis values seem to have changed from the previous version. It is unclear (i) why they have changed and (ii) why the x-axes now differ between panels (a) and (b). This needs to be reconciled.

*The changes in relation to the OVF (x-axis in Figure 9) were due to the removal of the NaSO4 hydrate component from the computation of fio, which had a subsequent impact on the OVF computed from volatility measurements. This component was removed because it has been shown that the contribution to volatility from NaSO4 hydrates is very small (Rasmussen et al. 2017).*

*The heated HGF (right hand side) figure previously displayed the total volatility on the x-axis, it now shows the low volatility OVF – because this is what contributes to the heated HGF. The x-axis has been relabelled.*

[Figure]

Figure 9

P21/L23: It is still not clear that this enhancement in droplet diameter is important at much higher RH than considered here. I suggest the authors make this aspect clear.

*The authors agree with the reviewer's statement and the paragraph has been changed to make this more clear.*

**P21/L23**
**"A buffered response of SSA hygroscopicity under supersaturated conditions to OVF has been previously reported (Ovadnevaite et al., 2011a; Collins et al., 2016; Forestieri et al., 2018b) and is thought to be linked to surface active organics (e.g. fatty acids). It should be noted that these observations were made in supersaturated conditions and therefore the applicability to observations in this study is not clear."**

P22/L9: It is unclear whether this organic fractions are for PM1 or for the 50 nm particles.

*The authors agree with the reviewer's statement and the paragraph has been changed to make this more clear.*

**P22/L6**
**"The 50 nm deliquescence relative humidity was measured for the Workboat 9 seawater sample at 69% RH (Fig. 10), notably lower than that observed for NaCl/sea salt, ~73.5 % (Zieger et al., 2017). SSA generated from Workboat 9 seawater displayed an ambient 50 nm HGF of 1.84 \pm 0.06, a heated 50 nm HGF of 1.94 \pm 0.06, an organic volume fraction of approximately 21% at 50 nm and a PM1 organic mass fraction of approximately 3\%. The PM1 alkane to hydroxyl ratio..."**

P23/L10: While I generally try to avoid bringing up new things during re-review, have the authors considered the sensitivity of their OCEANFILMS results to the assumed bubble thickness? They note in the methods that the assumed bubble thickness affects the resulting organic fraction. Can better agreement be obtained using a different, yet still reasonable, value for the bubble thickness? That said, the model/observation agreement is better at high OMF and worse at lower values, so changing the bubble thickness is likely to affect the intercept more than the slope, unless the authors were to allow the bubble thickness to vary with the water composition in some manner.

> *The bubble thickness will shift the values up or down, and as the reviewer said this could be used to improve the intercept, but not so the $R^2$. To optimise the intercept the thickness would need to be set to 0.8 - 1 um, which is at the upper limit of the bubble thickness in the literature (Modini et al. 2013 indicates a thickness of 0.01 to 1 um).*

> *It is likely that the thickness varies with organic fraction as the reviewer pointed out. For example measurements have shown that the presence of surfactants can enhance the bubble lifetime at the surface - allowing it to drain more and become thinner. However applying a parameterisation for the thickness is beyond the measurements taken in this study.*

> *This raises an interesting point and some discussion has been added on this.*

> **P23/L10**
> **"It is worth noting that both OCEANFILMS models over predict when the organic fraction is low, OMF < 0.05. The bias at low OMF could be due to the choice of bubble thickness. The use of a single thickness value is a simplification as it is likely to depend on the seawater surface composition and surface tension (Modini et al. 2013)."**

Compressed film model: I appreciate the updates that the authors have made. However, I still have concerns and questions over this discussion and implementation.

(i) For the ZSR comparison, the authors used the OVF values derived from the volatility measurements, specifically it seems that they used the total OVF. (Although it is not clear whether these are actually OMF or OVF…see question above regarding Fig. 9 discrepancy.) However, in the CF discussion, the authors use the OMF from the PM1 measurements, which they note on P26/L10. But why? This does not lead to a fair comparison, and makes it difficult to compare with Fig. S8, which shows the surface tension vs. the volatility derived OVF. Also, the OVF is, presumably, used as the input to the model, and thus seems to be a better parameter to compare with.

> *The authors agree with the reviewer's comments, the OVF is the more relevant parameter and is used as the input to the compressed film model. Figure 13/Figure S5 and the surrounding discussion now use OVF.*

[Figure]

Figure 13

(ii) Are the surface tensions reported in Fig. S8 the value at 90% RH? It is not clear.

*Yes the surface tensions are for 90%RH, this has been added to the caption.*

**"Surface tension computed using the compressed film model as a function of SSA organic volume fraction for 50 nm diameter SSA particles at 90% RH..."**

(iii) The authors state that "The reduction in surface tensions at low OVFs for the lipids plus polysaccharides and all organics partitioning cases results in over prediction of the HGF" and "the error in the modelled HGF is reduced at low OVFs when only the lipids are partitioned to the surface." It is not clear from Fig. 13 how this is the case. First, the use of absolute values of the measurement-model difference makes it not possible to tell what is an over vs. underestimate. Second, it is unclear what is being compared to what when it is stated that error is reduced. Looking at Fig. 13, the only difference between the panels is for the "functional groups" calculations for the lipid-only partitioning. The "OCEANFILMS" calculations for lipid-only are the same as for the total organic partitioning. Thus, how is error reduced comprehensively?

*This section has been reworded following adjustments of the molar volume in the compressed film model, and the associated changes to Figure 13.*

**P26/L8**
**"The most notable feature is that for both ambient (Fig 130 and heated measurements (Fig. S5) the error at high organic volume fractions is large. At high organic volume fractions the compressed film model performed best, at the assumed molecular volume and molecular area, when the lipids and proteins were allowed to partition to the surface. At low OVFs the predicted HGF is very similar regardless of which components were allowed to partition to the surface. At low OVFs the volume organics isn't sufficient for the**

**formation of a monolayer and therefore the surface tension remains equal to the surface tension of water, as shown in Fig. S8. The difference between the use of the functional group measurements or the OCEANFILMS-2 model for the functional group concentrations is distinguishable for the case in which only the lipids are allowed to partition to the SSA surface. In this case the higher lipid fraction predicted by OCEANFILMS-2 results in the formation of a monolayer at OVFs greater than 0.4, resulting in slightly lower error in the modelled HGFs. The monolayer is never formed for the lipid only case when the functional group measurements are applied, as indicated in Figure S8."**

(iv) The authors have selected to use a more reasonable organic molecular volume now. However, they assume this is appropriate for lipids. Given a typical lipid density of 1 g/cm3, a molecular volume of 4x10^-5 implies a molecular weight of 40 g/mol. This is inconsistent with the properties of lipids. Also, is there a reason why they have chosen to only consider the upper-limit organic HGF case? The data presented do not seem to support a large proportion of organosulfate species, which are the compounds that were observed to have such large HGF values. The lipopolysaccharides, which dominate the mass here (according to the FTIR measurements) have HGF values closer to 1.25 at 90%, per the cited Estillore et al. (2017) paper.

*The authors agree with the reviewers comments. A molecular volume of 2.2 * 10^-4 m3/mol. has now been applied, and this is now compared with an organic HGF of 1.15. This is consistent with the HGF from Raoult's law for an organic with a molecular volume of 2.2*10^-4 m3/mol. The methods section has been amended to describe the selection of molecular volume.*

**P12/L20**
**"The molecular volume was chosen to broadly represent the molecular volume of marine organics, most notably lipids and polysaccharides (Petters et al., 2009). The molecular area...'**

(v) Differences are seen between the functional group and OCEANFILMS model partitioning for lipid-only partitioning at low OVF. This is not discussed explicitly; instead, the authors focus on one of these results (from the functional group analysis) as this is the one result that differs from the others. A more complete discussion should be provided.

*The authors agree that this discussion was lacking. Discussion regarding the difference between the compressed film modeled HGFs using FTIR functional groups and OCEANFILMS-2 input has been included.*

**P26/L13**
**"The difference between the use of the functional group measurements or the OCEANFILMS-2 model for the functional group concentrations is distinguishable for the case in which only the lipids are allowed to partition to the SSA surface. In this case the higher lipid fraction predicted by OCEANFILMS-2 results in the formation of a monolayer at OVFs greater than 0.4 (greater than 0.5 for heated samples, Fig. S5), resulting in slightly lower error in the modelled HGFs. The monolayer is never formed for the lipid only case when the functional group measurements are applied, as indicated in Figure S8."**

(vi) It is not clarified in Fig. 13 that the results shown are likely for the functional group analysis, and not OCEANFILMS.

*Figure 13 includes data from both the functional groups and OCEANFILMS-2 compressed film model inputs. The authors suspect Figure 14 was being referred to. The figure caption in Figure 14 has been updated to explicitly state that the functional group data is being used in the compressed film modelled HGFs.*

**Figure 14. Ambient HGF modelled using compressed film model (green) as a function of OVF, modelled using the ZSR assumption (light blue) and observed (dark blue). Compressed film model output is for case where lipid and protein fraction (computed from functional group measurements) can partition to the particle surface. ZSR model used an organic HGF of 1.15.**

(vii) In Fig. 12 the authors show the absolute value of the measured minus modeled. At minimum, the y-axis label should be updated to indicate the absolute value has been used. However, by showing this as an absolute value, this masks the fact that at low OVF the modeled overpredicts the HGF and at higher OVF the model underpredicts. I suggest that this figure would be more meaningful if the true difference were shown, rather than the absolute value.

*The authors agree with the reviewers comments. Figure 13 and Figure S5 have been changed to show the difference between measured and modelled HGF.*

(viii) I am not certain that Fig. 14, as presented, "highlights the contribution of surface tension to the observed SSA HGF." The authors show results from the functional group analysis when only lipids partition. But, this is the case where surface tension has the smallest impact, per Fig. S8, since many points are unaffected. Only the points at the largest OVF are impacted by surface tension reduction. This is the reason that the ZSR and CF model results are so similar at low OVF. In particular, since the lipids are only ~10% of the total organics, this leaves 90% to contribute to the hygroscopicity with an apparent HGF of 1.6 (per the authors' constraints) and little influence on the surface tension. When the OVF is < 0.2, the fractional amount of organics that does not contribute to hygroscopicity is 0.2*0.1 = 0.02. At larger OVF, the majority of the organics still contribute to the hygroscopicity (since lipids are ~10%) but now the surface tension is reduced and there is slightly greater growth. The authors do note in the text that at high organic fraction the HGF is increased by 0.05 owing to the decrease in surface tension. I'll suggest that, if the authors' aim is to "highlight" the influence of surface tension, they add a panel that shows the difference between the ZSR and CF model results as a function of surface tension.

*The authors agree with the reviewer. Following the implementation of compressed film model parameters more consistent with marine organics, the discussion has now been amended*

**P26/L18**
**"A comparison of the HGFs modelled using the compressed film model and those modelled using the ZSR assumption is shown in Fig. 14. The compressed film model is shown for the case when the lipids and proteins are allowed to partition to the surface. The compressed film modelled HGFs at high organic volume fractions are up to 0.03 greater, however this does not account for the reduction in HGF predicted by the ZSR assumption, i.e. by Raoult's Law. The modest impact of decreased surface tension on HGF is consistent with previous studies on sub-saturated water uptake (Ruehl et al., 2016; Moore et al., 2008). Despite the inclusion of the surface tension effect (via the compressed film model) there was still a significant discrepancy between the observed and modelled HGFs, even when the relatively large uncertainty is the OVF is considered."**

(ix) I find the sentence on P26/L19 starting "The role of surface tension…" is a run-on. I think a period is missing after the first clause.

> *The authors agree and the first clause has been removed.*
> **P26/L28**
> **"OVFs inferred in studies using water uptake techniques, using the ZSR assumption (Fuentes et al., 2011; Modini et al., 2010a), have often been lower.."**

(x) Now that the authors CF calculations are clearer, I will suggest that the remainder of the paragraph starting on P26/L21 with "For example…" be substantially altered. The key point it seems the authors are trying to make here is that OVF values inferred from HGF measurements might lead to underestimates in the actual OVF. I think that the authors could better make this point by simply stating this, rather than trying to make a potentially apples-to-oranges comparison between different studies that took place at different times (which they seem to acknowledge in the last sentence). As currently written, the main point, I find, gets a little lost. I'll further encourage the authors to be semi quantitative here. They should be able to say, at least approximately, by how much the OVF might be underestimated. The magnitude of the differences in the HGF values shown in Fig. 14 b/t the ZSR and CF calculations suggest the potential mis-application of the ZSR could lead to a differences in the OVF of nominally 5% when the OVF is large (e.g. 0.75 instead of 0.8). I do not think that the authors' arguments or calculations would support differences of up to 40% in the OVF, as implied by the cited studies and numbers.

> *The authors agree with the reviewer's comments. An example of the average OVF computed using the ZSR assumption on the observed HGFs has been added to the text and compared to that computed using the volatility methods.*
>
> **Page 26/L25**
> **"Applying the ZSR assumption to the observed HGFs, assuming an organic HGF of 1.15, yields organic volume fractions for Bloom 1 of 0.12 pm 0.2 (mean _ sd), the OVFs computed using volatility measurements for Bloom 1 were 0.5 pm 0.3. It is also worth noting that the ZSR OVF for Bloom 1 includes an outlier (Workboat 4), for which the ZSR approximation is consistent with the OVF from volatility measurements. OVFs inferred in studies using water uptake techniques, using the ZSR assumption (Fuentes et al., 2011; Modini et al., 2010a), have often been lower than those measured using more direct analyses of SSA chemical composition (Keene et al., 2007; Facchini et al., 2008; Prather et al., 2013). SSA with hygroscopic growth factors inconsistent with the ZSR mixing rule, as presented here, could contribute to this apparent discrepancy."**

(xi) The authors note that the "discussion of the potential for surface tension to impact CCN concentrations has been removed. This section has been updated to discuss the potential difference between the CCN computed using ZSR modelled and measured HGFs." However, I still have concerns regarding this discussion. The authors implicitly assume that whatever phenomenon that is leading to the measured HGF here being relatively independent of the OVF persists to the point of activation. This is not guaranteed, and the authors seem to imply from the discussion previous to the CCN discussion that surface tension is responsible for at least some of the influence. I strongly encourage the authors to restrict their discussion to what they have measured (HGF values) or clearly state the assumption regarding translation of the HGF values measured at 90% to the critical supersaturation.

*The authors agree with the reviewers suggestions and have removed the discussion/calculations relating to the impact on CCN, instead the focus is on the sub saturated hygroscopicity.*

(xii) I suggest the authors report the lower-limit surface tension and the C0 value used in the main text. The lower-limit surface tension is important to the results.

*The authors agree and these details have been added to the data analysis section.*
**P12/L19**
**"A C0 value of 10^9 mol.mol^-1 and a surface tension minimum of 0.03 J/m^2 was applied."**

Minor:
P1L4: Should be "and therefore"
*The authors agree and the text has been changed* **"and therefore the region is..."**

P7/L18: Typically, Figures are numbered in the order they are presented. Thus, this should probably be Fig. 3.

*The reference to figure 4 has been removed from this section. It was unnecessary to refer to this figure (results) in the methods section.*

P5/L6: The Aitken mode composition was also estimated from the UFO-TDMA. This might be noted.
*The authors agree and the text has been changed*
"**Aitken mode SSA composition was inferred from volatility data and estimated from UFO-TDMA measurements. $PM_1$ composition.."**

P12/L35: There is extraneous text where I believe a reference should be.
*The authors have fixed the broken in text reference.*
P19/L25: This is an incomplete sentence.
*The authors agree and the text has been changed*
**"The ZSR assumption was used with an organic growth...."**

**Compare Results**

Old File:

**Cravigan_SOAP_NascentSSA_ReviewerComments.pdf**

**36 pages (3.14 MB)**
21/02/2020 4:54:08 PM

versus

New File:

**Cravigan_CopernicusTemplate_FinalSubmission.pdf**

**36 pages (1.24 MB)**
5/06/2020 8:02:05 PM

**Total Changes**

**573**

**Content**

97 Replacements
224 Insertions
176 Deletions

**Styling and Annotations**

10 Styling
66 Annotations

Go to First Change (page 1)

[revised manuscript text omitted]